# Deep Linear Network Training Dynamics from Random Initialization: Data, Width, Depth, and Hyperparameter Transfer

**Blake Bordelon** [1 2 3]  **Cengiz Pehlevan** [1 2 3]

## Abstract

We theoretically characterize gradient descent dynamics in deep linear networks trained at large width from random initialization and on large quantities of random data. Our theory captures the "wider is better" effect of mean-field/maximum-update parameterized networks as well as hyperparameter transfer effects, which can be contrasted with the neural-tangent parameterization where optimal learning rates shift with model width. We provide asymptotic descriptions of both non-residual and residual neural networks, the latter of which enables an infinite depth limit when branches are scaled as $1/\sqrt{\text{depth}}$. We also compare training with one-pass stochastic gradient descent to the dynamics when training data are repeated at each iteration. Lastly, we show that this model recovers the accelerated power law training dynamics for power law structured data in the rich regime observed in recent works.

## 1. Introduction

Understanding the training dynamics and scaling properties of neural networks is critical to deriving parameterizations and optimizers that achieve stable dynamics across model sizes (Yang et al., 2021; Bordelon et al., 2024c; Yang et al., 2023; Vankadara et al., 2024; Everett et al., 2024), which in turn lead to significant computational savings in hyperparameter search via optimal hyperparameter transfer from smaller to larger sized models (Achiam et al., 2023). However, characterizing the training dynamics of neural networks is generally complicated due to nonlinear dynamics that depend on the parameterization of the network, the weight initialization strategy, optimizer, and data structure.

[1]John Paulson School of Engineering and Applied Sciences, Harvard University [2]Center for Brain Sciences [3]Kempner Institute. Correspondence to: Blake Bordelon <blake_bordelon@g.harvard.edu>, Cengiz Pehlevan <cpehlevan@seas.harvard.edu>.

*Proceedings of the 42nd International Conference on Machine Learning*, Vancouver, Canada. PMLR 267, 2025. Copyright 2025 by the author(s).

Despite the complexity of the dynamics induced by gradient descent from random initial conditions, significant theoretical insight into deep network training can be gained by studying tractable limits (large width or large depth networks) or specialized architectures (such as deep linear networks) or the behavior of training under specific data distributions (such as random isotropic data). While the literature on various special cases of training contains numerous works which we review in Section 1.1 below, a theoretical model that can capture simultaneously the effect of varying width, depth, richness, data quantity on the learning dynamics in randomly initialized networks is missing. Of specific interest is that no existing theory can theoretically characterize the two effects necessary for successful hyperparameter transfer (Yang et al., 2021): (1) monotonic improvement in performance with model width or model depth and (2) approximately consistent optimal learning rates when appropriately parameterizing the model and optimizer.

***Is there a simple theoretical model of deep networks that captures the influence of random initialization, network parameterization, width, and depth in order to characterize the hyperparameter transfer effect?***

In this work, we provide such a theory, though for a limited model class of deep linear networks, by characterizing the training dynamics from random initialization on random data which captures both the effect of network width, depth, dataset size, and feature learning strength (richness). Concretely our contributions are

- We develop a novel dynamical mean field theory (DMFT) which is specific for deep linear neural networks that enables a prediction of typical case loss and representation trajectories over both initial weights and random data, or stochastic gradient descent (SGD) samples, as the disorder. This theory completely characterizes the train and test loss dynamics in terms of a closed set of equations for correlation and response functions at each hidden layer of the network.

- We show that our theory provides a minimal model that captures the impact of parameterization on training dynamics. In particular, our results capture both the "wider is better" effect of deep networks throughout

training as well as the hyperparameter transfer effect for networks in $\mu$P. Conversely, our theory captures that wider networks can train more slowly and hyperparameters can fail to transfer in NTK parameterization.

- We extend our theory to provide a solution to the training dynamics of deep *residual networks*, enabling the study of the large depth limit, though in a tractable setting. In this setting, we reproduce that hyperparameter transfer across depths by scaling the residual branch by $1/\sqrt{\text{depth}}$, consistent with prior works.

- We also show that our theory can capture the behavior of linear networks trained on structured data, such as power-law covariance eigenvalues. This results in power law scaling of the loss with training iterations, though with improved exponents compared to lazy learning, consistent with recent findings in the shallow case (Bordelon et al., 2024b).

## 1.1. Related Works

**Statics and Bayesian Networks**  Theory of the statics[1] of deep Bayesian linear neural networks in the regime where dataset $P$ and width $N$ are proportionally large were first characterized in Li & Sompolinsky (2021), resulting in a theory of a renormalized kernel after training. Pacelli et al. (2023) and Baglioni et al. (2024) theoretically and experimentally argue that the scale-renormalization of the NNGP kernel continues to capture the effect of feature learning in *nonlinear* Bayesian networks. Their results were recently extended to convolutional networks (Aiudi et al., 2025; Bassetti et al., 2024). Learning curves which average over a randomly sampled training set were obtained by Zavatone-Veth et al. (2022) with the replica method, allowing contrast between Bayesian inference over all layers with training random feature models, who extended the theory to structured covariates (Zavatone-Veth & Pehlevan, 2023). Cui et al. (2023a) showed that these learning curves also describe learning curves for *nonlinear* multilayer Perceptrons (MLPs) under Bayes-optimal teacher-student learning setup in the proportional limit. Hanin & Zlokapa (2024) used similar techniques to analyze weakly nonlinear Bayesian networks, where the nonlinearity is depth dependent, at large width $N$, depth $L$ and dataset size $P$ with $\frac{LP}{N}$ fixed.

**Dynamics from Random Initialization**  In the lazy learning regime, the infinite limit of a randomly initialized network is characterized by the Neural Tangent Kernel (NTK) (Jacot et al., 2018; Lee et al., 2019). However, if one adopts a scaling that enables feature learning at infinite width (mean field / $\mu$P scaling) then the dynamics of deep network training from random initialization are more complicated.

The limiting $N \to \infty$ gradient descent dynamics in MLPs for a fixed (finite) stream of data in the rich regime have been characterized with partial differential equations in one-hidden layer networks (Mei et al., 2019; Rotskoff & Vanden-Eijnden, 2022) as well as Tensor programs (Yang & Hu, 2021) and dynamical mean field theory (Bordelon & Pehlevan, 2022) techniques for deeper networks. These methods describe the typical case behavior of training nonlinear networks over random realizations of the initial weights. However, the resulting theories are generally intractable as they scale poorly with the size of the training dataset $P$ and the number of steps of training $T$ outside of the lazy training regime. A key challenge in this deep setting is to also average over the random data to obtain a reduced description where effects of order $\frac{P}{N}$ are preserved. Various perturbative approximations $\frac{1}{N}$ have been studied, though these do not capture a full joint limit for data and width and also suffer from intractability at large dataset sizes (Dyer & Gur-Ari, 2020; Roberts et al., 2022; Hanin & Nica, 2019; Bordelon & Pehlevan, 2023).

**Residual Networks and Large Depth Limits**  Theory to characterize infinite width and depth training dynamics $N \to \infty$ and $L \to \infty$ have been recently obtained for networks starting at random initial conditions Bordelon et al. (2024c); Yang et al. (2023). More recently, the work of Bordelon et al. (2024b) studied depth $L = 2$ finite width linear networks trained with a projected gradient descent using mean field techniques. In this present work, we characterize exact gradient descent for arbitrary depth $L$ networks. We also study the role of $\mu$P scaling to achieve learning rate transfer across width and scaled residual branches (Bordelon et al., 2024a; Yang et al., 2023) for transfer across depths.

**Linear Network Gradient Flow ODEs**  The dynamics of gradient flow training in deep linear networks in the very rich regime (or equivalently for small initialization) were computed by Saxe et al. (2013); Atanasov et al. (2022) as a simple set of ordinary differential equations. Kunin et al. (2024) and Dominé et al. (2024) recently derived similar equations for the dynamics of networks at different laziness/richness scale and unbalanced initializations. However, these methods require a number of assumptions including whitened input data, a balance assumption on products of weights in adjacent layers, and lack of bottleneck or overparameterized layers. Our work goes beyond these assumptions by studying generic (random and non-negligible) initialization of the network weights, which introduce interesting dependence of the dynamics on network width and dataset size. In addition, our theory allows for sampling random training data, which will generally not give a whitened empirical covariance[2].

---

[1]This could be interpreted as sampling a neural network from an equilibrium distribution, which can be contrasted with out-of-equilibrium dynamics which are our focus in this work.

[2]Even if the data's population distribution is isotropic, the empirical eigenvalues will follow a Marchenko-Pastur law.

**Linear Network Models of Neural Scaling Laws.** Bordelon et al. (2024a) and Paquette et al. (2024) recently analyzed a linear random feature model trained with gradient descent, finding that training dynamics on data that satisfies source and capacity conditions recovers Chinchilla-like neural scaling laws (Hoffmann et al., 2022). Bordelon et al. (2024b) recently extended this theory to a case where the random projection matrix is also updated with projected gradient descent, resulting in a faster power law convergence rate for hard tasks. Their model is mathematically equivalent to a one hidden layer linear network trained with a projected version gradient descent. We find this same acceleration behavior in *deeper* linear networks trained on power law data in this work.

**DMFT Approaches to Machine Learning Theory** Dynamical mean field theory methods, originally developed in the physics of spin glasses, have been used to describe the dynamics of high dimensional optimization (including stochastic gradient methods) in terms of single variable stochastic integro-differential equations (Agoritsas et al., 2018; Mannelli et al., 2019; Mignacco et al., 2020; Mignacco & Urbani, 2022; Gerbelot et al., 2022; Dandi et al., 2023; Bordelon et al., 2024a;b). These prior works have shown the validity of DMFT to characterizing general linear models or shallow neural networks trained with parameters and data going to infinity proportionally. In our work the disorder comes from *both the data matrix and the random initial weights* of the network and we are interested in the behavior of arbitrary depth networks.

## 2. Deep Linear Network with Random Data

The first model that we study is a vanilla feedforward linear MLP which has $L$ hidden layers of neurons (and $L + 1$ trainable weights) which maps input $\boldsymbol{x} \in \mathbb{R}^D$ to outputs $f(\boldsymbol{x})$ defined as

$$f(\boldsymbol{x}) = \frac{\sqrt{D}}{N\gamma_0} \left(\boldsymbol{w}^L\right)^\top \prod_{\ell=1}^{L-1} \left(\frac{1}{\sqrt{N}}\boldsymbol{W}^\ell\right)\left(\frac{1}{\sqrt{D}}\boldsymbol{W}^0\right)\boldsymbol{x}. \tag{1}$$

The first matrix has dimensions $\boldsymbol{W}^0 \in \mathbb{R}^{N \times D}$, while all intermediate weight matrices are all of size $\boldsymbol{W}^\ell \in \mathbb{R}^{N \times N}$. Lastly, the output $\boldsymbol{w}^L \in \mathbb{R}^N$ maps to a single output. We can provide extensions to unmatched hidden widths, structured data, and multiple (finite) output classes in Appendix F, E, and G, but we begin our analysis with this simplest possible case. We start by characterizing the dynamics for a random training dataset $\mathcal{D} = \{\boldsymbol{x}_\mu, y_\mu\}_{\mu=1}^P$ of size $P$, and random initial weights

$$\boldsymbol{x}_\mu \sim \mathcal{N}(0, \boldsymbol{I}) \, , \, w_i^L, W_{ij}^\ell, W_{ij}^0 \sim \mathcal{N}(0, 1). \tag{2}$$

We consider a linear (potentially noisy) target function of the form

$$y(\boldsymbol{x}) = \frac{1}{\sqrt{D}}\boldsymbol{w}_\star \cdot \boldsymbol{x} + \sigma\epsilon(\boldsymbol{x}) \tag{3}$$

where $\langle\epsilon(\boldsymbol{x})\rangle = 0$ and $\langle\epsilon(\boldsymbol{x})^2\rangle = 1$. All parameters follow gradient descent dynamics on the square loss with the $\mu$P scaling of the update from gradient descent which takes the form

$$\boldsymbol{W}^\ell(t+1) = \boldsymbol{W}^\ell(t) + \frac{\eta\gamma_0}{\sqrt{N}}\,\boldsymbol{g}^{\ell+1}(t)\boldsymbol{h}^\ell(t)^\top, \tag{4}$$

where we introduced the following vectors

$$\boldsymbol{h}^{\ell+1}(t) = \frac{1}{\sqrt{N}}\boldsymbol{W}^\ell(t)\boldsymbol{h}^\ell(t) \, , \, \boldsymbol{h}^0(t) = \frac{\sqrt{D}}{P}\sum_{\mu=1}^P \Delta_\mu(t)\boldsymbol{x}_\mu$$

$$\boldsymbol{g}^\ell(t) = \frac{1}{\sqrt{N}}\boldsymbol{W}^\ell(t)^\top\boldsymbol{g}^{\ell+1}(t) \, , \, \boldsymbol{g}^L(t) = \boldsymbol{w}^L(t). \tag{5}$$

and the instantaneous training errors $\Delta_\mu(t) = y(\boldsymbol{x}_\mu) - f(\boldsymbol{x}_\mu, t)$. We note that the vectors $\boldsymbol{h}^\ell(t)$ defined above differ from the preactivations used in the DMFT of Bordelon & Pehlevan (2022) since ours are *training-set averaged* quantities which are computed as a correlation between the errors $\Delta_\mu$ and the usual per-sample preactivation vectors. The fact that sums over training points and sums over neurons can be reordered in linear networks enables easy analysis of a proportional limit, where width, data, and input dimension are all comparable, rather than only computing the $N \to \infty$ limit at fixed $P, D$. In addition to capturing better finite width and finite data effects, this theory leads to significantly simpler and easier-to-compute description of training dynamics as we illustrate in Appendix Table 1.

### 2.1. Dynamics of (S)GD in a Joint Asymptotic Limit

In this work we consider two distinct setting that allow us to identify different effects from subsampling data. First, we examine full batch gradient descent with $P$ data points that are reused at each step. In this setting, we study a large system size limit with $P, D, N \to \infty$ limit where the ratios are constants

Setting 1 (Full Batch GD): $\quad P/D \equiv \alpha \quad , \quad N/D \equiv \nu$ (6)

Analyzing fixed points of GD in the above limit is a popular setting for two layer random feature models (Mei & Montanari, 2022; Adlam & Pennington, 2020a; Hu & Lu, 2022) and deep Bayesian networks at equilibrium (Li & Sompolinsky, 2021; Zavatone-Veth et al., 2022; Cui et al., 2023a) but the training dynamics from a random initial condition in such a limit has not been explored[3]. In addition to full batch

---

[3]The $\nu \to \infty$ limit would recover to the commonly studied "infinite width limits" where width is taken to infinity first at fixed $P, D$ that disregards effects of order $N/D$ or $N/P$ (Jacot et al., 2018; Lee et al., 2019; Bordelon & Pehlevan, 2022).

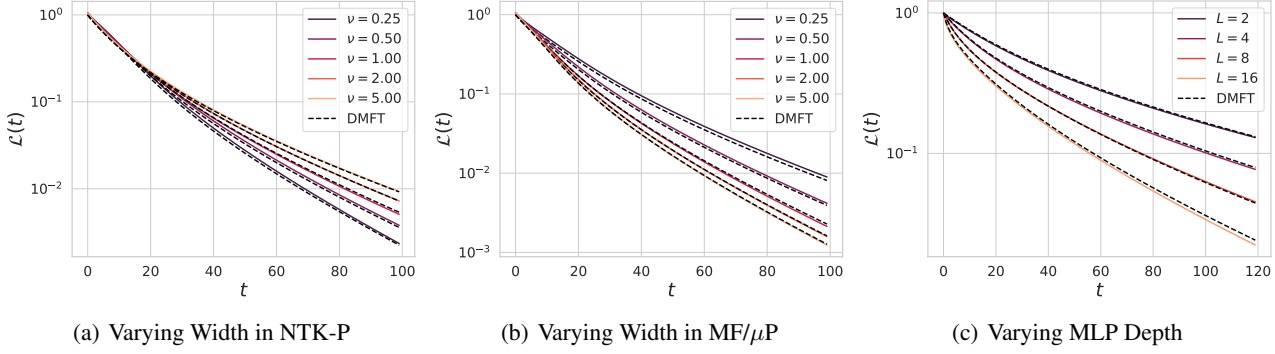

| (a) Varying Width in NTK-P | (b) Varying Width in MF/$\mu$P | (c) Varying MLP Depth |

*Figure 1.* Our theory captures the role of width, depth and dataset size on the test loss $\mathcal{L}(t)$ and train loss $\hat{\mathcal{L}}(t)$ dynamics. The behavior of different models across widths depends on how $\gamma_0$ scales with $N$. (a) Test loss of a $L = 4$ network in NTK parameterization where $\gamma_0 \sim \Theta(\frac{1}{\sqrt{\nu}})$, which results in a kernel method (linear model) in the $\nu \to \infty$ limit. (b) In MFP/$\mu$P, the parameter $\gamma_0 = \Theta(1)$ and the convergence rate improves as $\nu$ increases. (c) Our theory allows us to track dynamics in deeper networks of arbitrary depth $L$ as well.

gradient descent training, we also look at online stochastic gradient descent (SGD) training, where at each step a fresh batch of $B$ data points are sampled and used to estimate the gradient with the following proportional limit

Setting 2 (Online SGD):   $B/D \equiv \alpha_B$  ,   $N/D \equiv \nu.$
$$(7)$$

These two settings (full batch GD and online SGD) both converge to the dynamics of gradient descent on the population loss as $\alpha \to \infty$ or $\alpha_B \to \infty$ respectively. In either of these settings, large system size limits are taken with the number of iterations $t$ fixed, which is appropriate since all learning timescales are $\mathcal{O}(1)$.

The key quantities which determine the generalization dynamics in both of our settings are correlation functions

$$C_h^\ell(t,t') = \frac{1}{N} \boldsymbol{h}^\ell(t) \cdot \boldsymbol{h}^\ell(t') \,, \; C_g^\ell(t,t') = \frac{1}{N} \boldsymbol{g}^\ell(t) \cdot \boldsymbol{g}^\ell(t')$$
$$C_\Delta(t,t') = \frac{1}{P} \boldsymbol{\Delta}(t) \cdot \boldsymbol{\Delta}(t') \,, \; C_v(t,t') = \frac{1}{D} \boldsymbol{v}(t) \cdot \boldsymbol{v}(t'),$$
$$(8)$$

where the error vector $\boldsymbol{v}(t)$ is defined as

$$\boldsymbol{v}(t) = \boldsymbol{w}^\star - \frac{\sqrt{D}}{N\gamma_0} \boldsymbol{W}^0(t)^\top \boldsymbol{g}^1(t). \qquad (9)$$

We stress that the number of these order parameters **does not depend on** $D, N, P$ but only scales with the number of iterations $T$ and the number of layers $L$, which justifies the saddle point derivation in the Appendix A. From the correlation functions, we can directly obtain the test loss $\mathcal{L}(t)$ and train loss $\hat{\mathcal{L}}(t)$ as

$$\mathcal{L}(t) = C_v(t,t) + \sigma^2 \,, \; \hat{\mathcal{L}}(t) = C_\Delta(t,t). \qquad (10)$$

In addition to the correlation functions, the theory also requires computing a collection of response functions $\{R_{vu}^0, R_{vr}^0, R_\Delta, R_{hr}^0, ... R_{hr}^{L-1}, R_{gu}^1, ..., R_{gu}^L\}$ which capture the effect of limited data and limited width from finite $\alpha$ and finite $\nu$ effects respectively as well as interactions between the forward/backward $\boldsymbol{h}^\ell, \boldsymbol{g}^\ell$ variables (see Appendix A). In the joint limit, each entry of the vectors $\{\boldsymbol{v}(t), \boldsymbol{\Delta}(t), \boldsymbol{h}^\ell(t), \boldsymbol{g}^\ell(t)\}$ become statistically independent and identically distributed following a stochastic process known as the single-site process. As an example, the intermediate layers of the network $1 < \ell < L$ satisfy the following single site (scalar) processes

$$h^\ell(t) = u^\ell(t) + \sum_{t'<t} \left[ R_{hr}^{\ell-1}(t,t') + \eta\gamma_0 C_h^{\ell-1}(t,t') \right] g^\ell(t')$$
$$g^\ell(t) = r^\ell(t) + \sum_{t'<t} \left[ R_{gu}^{\ell+1}(t,t') + \eta\gamma_0 C_g^{\ell+1}(t,t') \right] h^\ell(t')$$
$$(11)$$

where $u^\ell(t) \sim \mathcal{N}(0, C_h^{\ell-1})$ and $r^\ell(t) \sim \mathcal{N}(0, G^{\ell+1})$ are colored noise sources. The response functions are computed as derivatives of these single site variables with respect to their noise source

$$R_{hr}^\ell(t,t') = \left\langle \frac{\delta h(t)}{\delta r^\ell(t')} \right\rangle \,, \; R_{gu}^\ell(t,t') = \left\langle \frac{\delta g^\ell(t)}{\delta u^\ell(t')} \right\rangle. \qquad (12)$$

Intuitively, these functions measure the response of the variables at time $t$ to a "kick" in the noise source at a time $t'$ (due to causality, these only contribute for $t' < t$). We provide the complete set of DMFT equations in Appendix A.2, which form a closed set of equations for the correlation and response functions. Once these are solved for, the the test

error takes the form

$$\mathcal{L}(t) = \underbrace{\sum_{s<t}\sum_{s'<t} R_{vr}^0(t,s)R_{vr}^0(t,s')}_{\text{Bias}} + \underbrace{\sigma^2}_{\text{Label Noise}}$$

$$+ \underbrace{\frac{1}{\nu\gamma_0^2}\sum_{s<t}\sum_{s'<t} R_{vr}^0(t,s)R_{vr}^0(t,s')C_g^1(s,s')}_{\text{Limited Width Variance}}$$

$$+ \underbrace{\frac{1}{\alpha}\sum_{s<t}\sum_{s'<t} R_{vu}^0(t,s)R_{vu}^0(t,s')C_\Delta(s,s')}_{\text{Limited Data Variance}} \quad (13)$$

The last two lines capture the fluctuations in the entries of the error vector $v(t)$ which are due to limited width (finite $\nu$) and limited data (finite $\alpha$) respectively while the first line would be the error dynamics of an ensembled and bagged model (the error of the average function learned over different initializations and different draws of training data) (Adlam & Pennington, 2020b; Bordelon et al., 2024a). The response function $R_{vr}^0$ also depends on both $\nu$ and $\alpha$ and captures the effect of finite $\nu$ and $\alpha$ on the bias dynamics, which persist even for an ensembled and bagged predictor.

**Varying Width and Parameterization**   Our theory captures the effect of varying width through the constant $\nu \equiv N/D$. Depending on how the model is parameterized with respect to model width $N$, different behaviors are possible as $\nu$ varies in the proportional limit. In Figure 1 (a)-(b) we compare how models behave across different $\nu$ in NTK parameterization (where $\gamma_0 \sim \Theta\left(\nu^{-1/2}\right)$) compared to mean field parameterization where $\gamma_0$ is constant with respect to $\nu$. In NTK parameterization, the network approaches a linear model (in parameter space) as $\nu \to \infty$, whereas the dynamics remain non-linear in the $\nu \to \infty$ limit. In either case, the key equation which captures the effect of network width is the single-site equation for the firs layer $h^1(t)$ variables

$$h^1(t) = \underbrace{u^1(t)}_{\text{Gaussian Process}} + \underbrace{\frac{1}{\nu\gamma_0}\sum_{t'<t} R_{hr}^0(t,t')g^1(t')}_{\text{Limited Width Effects}}$$

$$+ \underbrace{\eta\gamma_0\sum_{t'<t} C_h^0(t,t')g^1(t')}_{\text{Feature Learning Updates}}. \quad (14)$$

These limited width effects can propagate through the other layers in the forward and backward pass and ultimately bias the dynamics of the network predictions, leading to slower training. For mean field parameterization, the feature learning updates scale as $\sim \eta\gamma_0$ which is constant across model widths, while for NTK parameterization, the feature learning update scales as $\sim \eta\gamma/\sqrt{\nu}$ which decrease with model width.

**Learning Rate Transfer**   Experiments from Yang et al. (2021) illustrate that the optimal learning rate is approximately preserved across model scales in the $\mu$P scaling, while Noci et al. (2024) show experiments where optimal learning rates shift to the right as width increases in NTK parameterization. Our theory can capture both of these effects as we show in Figure 2. The key effect is that, at fixed learning rate $\eta$, the rate of change of the kernel decays with network width $\nu$ in NTK parameterization, while it is approximately constant in $\mu$P.

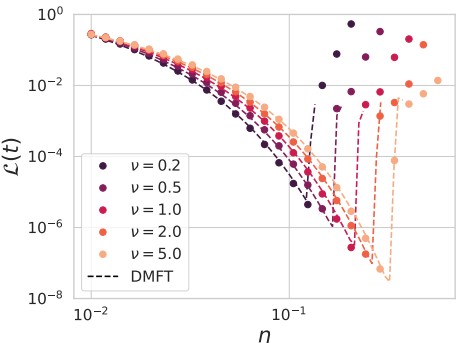

(a) Learning Rate Sweep in NTK-P

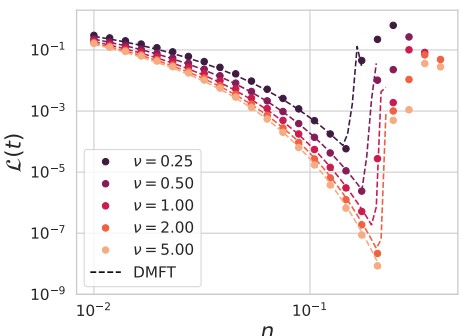

(b) Learning Rate Sweep in $\mu$P

*Figure 2.* The theory can capture the failure/success of learning rate transfer in NTK/$\mu$P networks. (a) The optimal learning rate increases with model width $\nu = N/D$ in NTK parameterization. This is due to the fact that the effective feature learning rate $\gamma_0$ is decreasing as $1/\sqrt{\nu}$. (b) The optimal learning rate is approximately preserved across $\nu = N/D$ in $\mu$P networks where $\gamma_0$ is constant, enabling learning rate transfer.

**Effect of Limited Data or Batch Size**   The effect of limited data on the dynamics generates a memory term for the training error $\Delta(t)$ response function

$$R_\Delta(t,t') = \delta(t-t') + \underbrace{\frac{1}{\alpha}\sum_{t''<t} R_{vu}^0(t,t'')R_\Delta(t'',t')}_{\text{Limited Data Bias}},$$

$$(15)$$

which results in a slow buildup in the gap between train and test loss dynamics as we illustrate in in Figure 3 (a). When

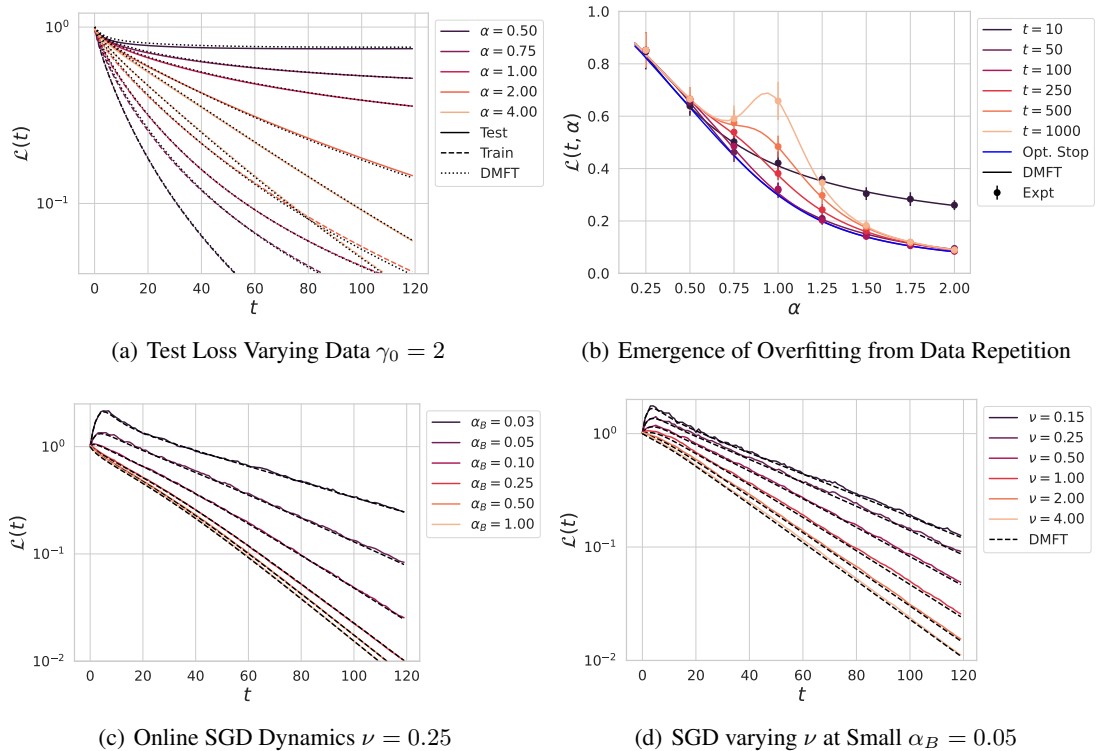

(a) Test Loss Varying Data $\gamma_0 = 2$

(b) Emergence of Overfitting from Data Repetition

(c) Online SGD Dynamics $\nu = 0.25$

(d) SGD varying $\nu$ at Small $\alpha_B = 0.05$

*Figure 3.* Train and test losses of a depth $L = 4$ linear network versus dataset size $P = \alpha D$ compared to online SGD with batch size $B = \alpha_B D$. (a) Networks can fit the training data faster for smaller datasets $\alpha$. (b) The test error can be non-monotonic as a function of data $\alpha$ if training is performed too long. (c) The dynamics of online SGD with $\nu = 0.5$ across batchsizes $B = \alpha_B D$. Smaller batchsizes slow down training through accumulated SGD variance, but there is not a bottleneck error as $t \to \infty$ provided the dynamics are stable. (d) Increasing the width $\nu$ eliminates the initial overfitting transient (see Appendix D.2).

data is repeated for many iterations, the loss can exhibit overfitting peaks (d'Ascoli et al., 2020; Mei & Montanari, 2022; Advani et al., 2020; Zavatone-Veth et al., 2022) as we show in Figure 3. However, for online SGD training the loss is monotonically improving with model width $\nu$ and batchsize $B = \alpha_B D$ as we illustrate in Figure 3 (c)-(d).

## 3. Residual Networks and Depth Limits

While we studied the large width limit at any fixed finite value of $L$, to obtain a well defined large depth $L \to \infty$ limit, we next turn to residual networks which improve faithful forward and backward signal propagation

$$f(\boldsymbol{x}) = \frac{\sqrt{D}}{N\gamma_0} \left(\boldsymbol{w}^L\right)^\top \prod_{\ell=1}^{L-1} \left(\boldsymbol{I} + \frac{\beta}{\sqrt{N}} \boldsymbol{W}^\ell\right) \left(\frac{1}{\sqrt{D}} \boldsymbol{W}^0\right) \boldsymbol{x}$$

(16)

We first provide the DMFT for the proportional $P, D, N$ limit for any finite $\beta$ and $L$. We show that unlike standard feedforward networks, residual networks require characterizing cross-layer correlation and response functions since the residual stream variables are statistically dependent. Once we have derived the proportional limit, we can then consider

scaling $\beta = \beta_0/\sqrt{L}$ for constant $\beta_0$, which recovers the parameterization recovers the depth-$\mu$P of (Bordelon et al., 2024c; Yang et al., 2023)[4].

We show our theory compared to experiments in Figure 4 where we illustrate the impact of richness $\gamma_0$, branch multiplier $\beta_0$ and depth $L$ on the dynamics when we are adopting the $\beta = \frac{\beta_0}{\sqrt{L}}$ scaling that admits a large depth limit. Examples illustrating hyperparameter transfer are provided in Figure 5, where keeping $\beta = \Theta(1)$ constant leads to diverging activity at fixed learning rate as $L \to \infty$ whereas scaling $\beta = \Theta(L^{-1/2})$ admits a well defined limit. Thus hyperparameter transfer in the rich regime is maintained in the $\beta = \beta_0/\sqrt{L}$ scaling.

**Infinite Depth Limit**   When scaling the branch multiplier as $\beta = \frac{\beta_0}{\sqrt{L}}$, the dynamics admit a $L \to \infty$ limit. This $L \to \infty$ limit is a simplified version of the result obtained in Bordelon et al. (2024c) where both the response functions and correlation functions survive in the $L \to \infty$ limit (see

---

[4]Alternatively, if one commits to $\beta = \beta_0/\sqrt{L}$, then one can take the $L \to \infty$ limit first and then take the proportional $N, P, D \to \infty$ limit, which gives the same result.

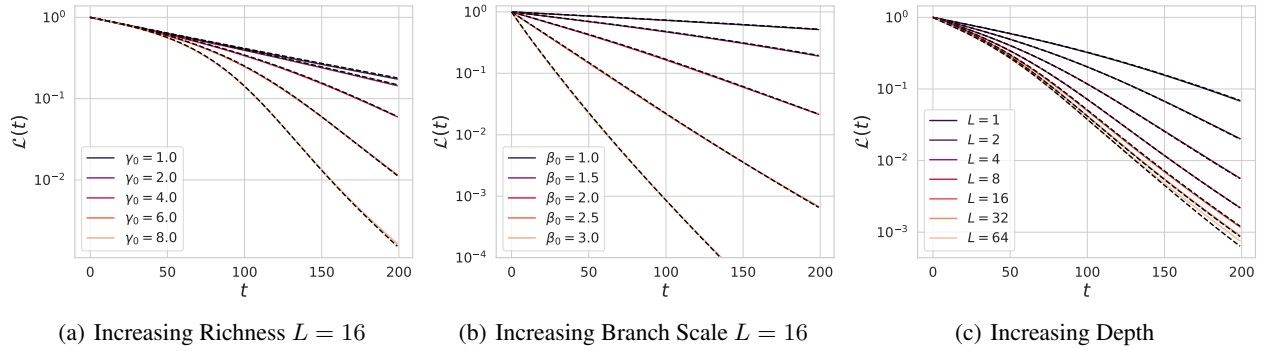

(a) Increasing Richness $L = 16$     (b) Increasing Branch Scale $L = 16$     (c) Increasing Depth

*Figure 4.* The training dynamics of large depth residual networks with random initialization. (a) Increasing $\gamma_0$ leads to an acceleration of training, but the initial kernel governing the updates are not changed so that $\frac{d}{dt}\mathcal{L}(t)|_{t=0}$ is preserved. (b) Increasing $\beta_0$ alters both the initial kernel and the feature learning dynamics. (c) Increasing depth $L$ with $\beta_0 = 1.2$ eventually converges to a limiting dynamics.

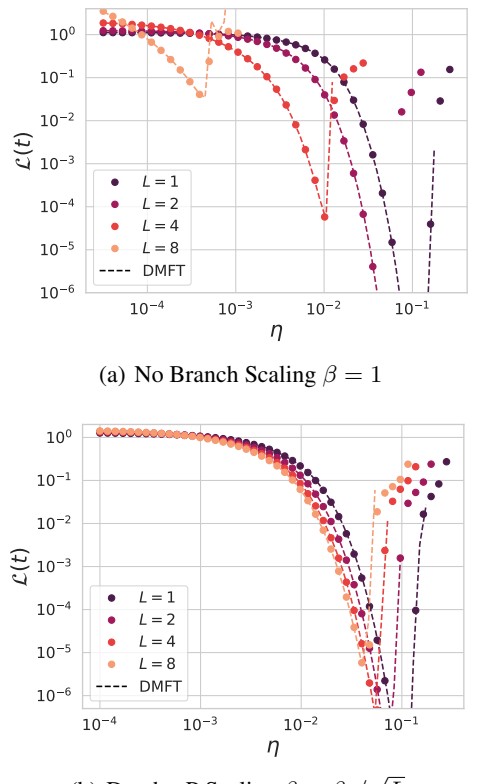

(a) No Branch Scaling $\beta = 1$

(b) Depth $\mu$P Scaling $\beta = \beta_0/\sqrt{L}$

*Figure 5.* Learning rate transfer fails for vanilla ResNets but succeeds for scaled resnets, consistent with theory (dashed lines). (a) ResNets of varying depth $L$ with constant raw branch scale $\beta = 1$. The optimal learning rate changes by several orders of magnitude. (b) ResNets with scaled branch $\beta = \frac{\beta_0}{\sqrt{L}}$ with $\beta_0 = 1.25$. The optimal learning rate does not shift significantly as depth increases.

Appendix C.1). In this case, the residual stream can be described as a stochastic process over layer time $\tau = \ell/L$ driven by a Brownian motion term $du(t, \tau)$ and a feature learning correction that depends on correlation and response

dynamics (to be integrated from $\tau = 0$ up to $\tau = 1$), giving single site dynamics

$$
dh(\tau, t) = \beta_0 du(\tau, t)
$$
$$
+ d\tau \beta_0^2 \sum_{t' < t} \left[ R_{hr}(\tau, t, t') + \eta \gamma_0 C_h(\tau, t, t') \right] g(\tau, t'). \tag{17}
$$

where the Brownian motion has covariance structure $\langle du(\tau, t) du(\tau', t') \rangle = \delta(\tau - \tau') C_h(\tau, t, t') d\tau d\tau'$ which is local in layer time $\tau$ but non-local across training steps (see Appendix C.1 for more detail).

## 4. Structured Data & Power Law Scalings

While the focus in the paper so far has been on data drawn from an isotropic distribution and on the proportional limit, we stress that our methods can also characterize the dynamics of training on structured data. For example, suppose that our covariates $\boldsymbol{x}$ have the following covariance structure

$$
\langle x_k x_\ell \rangle = \lambda_k \delta_{k\ell} \, , \ y(\boldsymbol{x}) = \sum_k w_k^\star x_k \tag{18}
$$

where the target weights $w_k^\star$ are arbitrary. Of particular interest are trace class covariances with $\sum_{k=1}^\infty \lambda_k < \infty$ which require developing a dimension-free version of the theory (Cheng & Montanari, 2022; Bordelon et al., 2024a). We develop an approximate mean field theory using the same techniques. The error along the $k$-th eigenspace of the data $v_k(t) = w_k^\star - \frac{1}{N\gamma_0} \boldsymbol{W}^0(t)^\top \boldsymbol{g}^1(t)$ follows the dynamics

$$
v_k(t) = \sum_{t' < t} \mathcal{H}_k(t, t') \left[ w_k^\star - r_k^0(t') - \tilde{u}_k^0(t') \right], \tag{19}
$$

where the Gaussian noise variables $r_k^0(t)$ and $\tilde{u}_k^0(t)$ are due to finite width effects and finite dataset (or finite batch) respectively. We introduced the transfer function $\mathcal{H}_k$ for each eigenmode $k$ that describes the (biased) trajectory of $v_k$'s dynamics in the absence of these noise terms (see Appendix

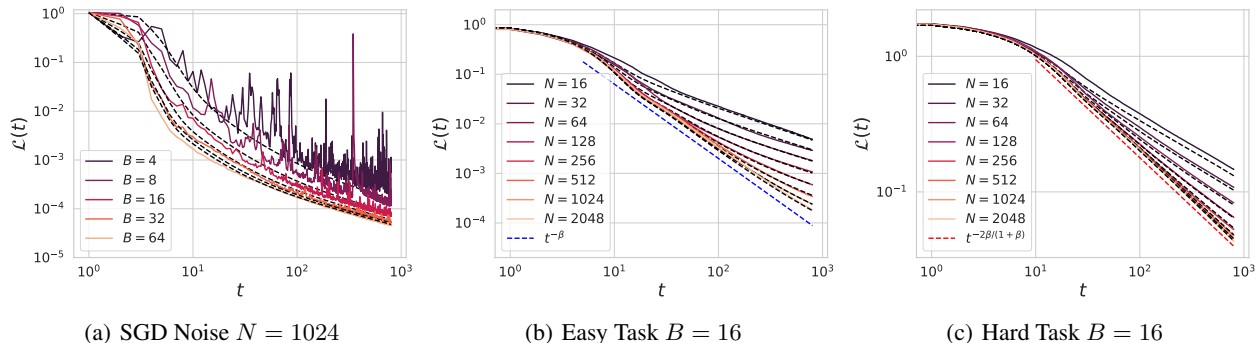

(a) SGD Noise $N = 1024$      (b) Easy Task $B = 16$      (c) Hard Task $B = 16$

*Figure 6.* A dimension free mean field description of the dynamics captures SGD effects and finite width $N$ effects for power law data covariates. (a) SGD with small batchsize $B$ is approximately captured by the mean field theory for $L = 4$. (b) Deep linear networks ($L = 4$) trained on "easy" power law tasks with $\beta > 1$ exhibit the same scaling law as in the lazy learning regime. Finite width networks eventually deviate from the infinite width limit with a slower convergence rate. (c) Deep linear networks trained on "hard" powerlaw tasks in the sense of (Bordelon et al., 2024b) exhibit a speedup of their scaling exponent relative to their convergence rate in the lazy limit.

E). This transfer function $\mathcal{H}_k(t)$ contains finite width $N$ and finite data $P$ effects (but not finite batch effects for SGD). The test loss can be computed as $\mathcal{L}(t) = \sum_k \lambda_k \langle v_k(t)^2 \rangle$. Of particular interest is the behavior of this model under powerlaw covariates

$$\lambda_k \sim k^{-\alpha} \ , \ (w_k^\star)^2 \lambda_k \sim k^{-\alpha\beta-1}, \qquad (20)$$

known as source/capacity conditions (Cui et al., 2023b; Bordelon et al., 2024a)[5]. Tasks with larger (smaller) $\beta$ are easier (harder) to estimate for the model. We illustrate the behavior of our theory for deep linear networks trained with SGD on these power law tasks compared to simulations in Figure 6. Despite this mean field description being only approximate for small $N, B$, the theory begins to become accurate around $N, B \sim 16$ for power law eigenspectra. While a linear model would always follow a power law scaling with training iterations $\mathcal{L} \sim t^{-\beta}$, we note that large width networks in the feature learning regime can obey the following power law scaling

$$\lim_{N \to \infty} \mathcal{L}(t, N) \sim \begin{cases} t^{-\frac{2\beta}{1+\beta}} & \beta < 1 \quad \text{(Hard Tasks)} \\ t^{-\beta} & \beta \geq 1 \quad \text{(Easy Tasks)} \end{cases} \qquad (21)$$

which agrees with the exponents derived by Bordelon et al. (2024b) in the one-hidden layer $L = 1$ case. However, our theory shows that this improved exponent for the hard task regime due to non-lazy dynamics is in fact the same for deeper networks as well. This indicates that the structure of the task could induce some universal behavior in linear networks of different architectures. We stress that this scaling law result is derived in the gradient flow limit of our more general DMFT, but edge of stability (EoS) effects could spoil this behavior at long enough training times in discrete

time (Ren et al., 2025; Cohen et al., 2021). While this effect should in principle be availabel from our existing DMFT equations, we leave a more detailed analysis of this effect in this model to future work.

**LR Transfer For Dimension Free Power Law Features**
While our learning rate transfer plots exhibited very sharp dependence on learning rate for isotropic features (Figure 2), we can develop more realistic hyperparameter transfer curves under our power law feature setting. In this dimension-free setting ($D \to \infty$ with power law exponent $\alpha > 1$), the theory depends explicitly on $N, B \gg 1$. We illustrate that our main findings regarding learning rate transfer persist for power law features in Figure 7.

## Discussion

In this work, we developed an asymptotic mean field description of the training dynamics of randomly initialized deep linear networks trained on random data. The theory was able to capture both vanilla MLP architectures and deep residual neural networks and characterize the impact of parameterization, learning rate, richness parameter, width, depth, dataset size, and batchsize for online SGD. This is the first work to also numerically verify the asymptotic description of training dynamics in infinite width residual networks. We demonstrated that our theory can capture the hyperparameter transfer effect for properly parameterized networks across both widths and depths which have been observed empirically prior works. For structured and trace-class covariates, we modify our theory to approximately capture the impact of finite batchsize $B$ and finite width $N$.

**Limitations and Future Directions** The current theory, while a signfcant improvement in complexity compared to prior works which do not average over the random data (such as Bordelon & Pehlevan (2022); Bordelon et al. (2024c)), becomes costly to compute for very deep networks

---

[5]Not to be confused with the $\alpha, \beta$ of previous sections. We use this to be consistent with the notation of (Bordelon et al., 2024b).

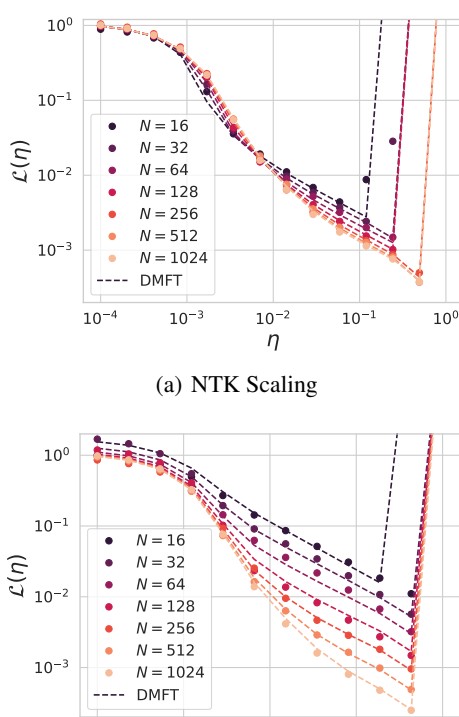

(a) NTK Scaling

(b) $\mu$P Scaling

Figure 7. Learning rate transfer experiments for a $L = 3$ $\mu$P linear network trained with SGD ($B = 128$) on powerlaw features $(\alpha, \beta) = (2, 1.75)$ and $\gamma_0 = 0.4$. (a) The hyperparameter transfer effect fails for NTK parameterization but (b) succeeds for $\mu$P and is accurately captured by DMFT (dashed).

## Impact Statement

This paper presents work whose goal is to advance the field of Machine Learning. There are many potential societal consequences of our work, none which we feel must be specifically highlighted here.

## Acknowledgements

We thank Boris Hanin, Mufan (Bill) Li, Lorenzo Noci, Jacob Zavatone-Veth, Hugo Cui, Francesca Mignacco, Stefano Sarao Manelli, and Clarissa Lauditi for useful and engaging conversations. BB thanks Jason Lee for pointing out that discrete time effects in our power law model would alter the new scaling law behavior on hard tasks after sufficient training time.

B.B. is supported by a Google PhD Fellowship. C.P. is supported by NSF grant DMS-2134157, NSF CAREER Award IIS-2239780, and a Sloan Research Fellowship. This work has been made possible in part by a gift from the Chan Zuckerberg Initiative Foundation to establish the Kempner Institute for the Study of Natural and Artificial Intelligence.

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

## A. Derivation of Mean Field Equations

In this section, we provide a mathematical derivation of the DMFT equations that govern the train and test loss dynamics of the deep linear network. We will start by analyzing the non-residual architecture trained with full batch gradient descent and then proceed to study online SGD and the residual network architecture, which involves a larger set of cross-layer response functions in the residual stream.

### A.1. Non-Residual Deep Linear Network Trained with Full Batch GD

#### A.1.1. ISOLATING DEPENDENCE ON THE DISORDER

We start by considering the feedforward depth $L$ network with no residual skip connections defined as

$$f(\boldsymbol{x}) = \frac{\sqrt{D}}{N\gamma_0}(\boldsymbol{w}^L)^\top \prod_{\ell=1}^{L-1} \left(\frac{1}{\sqrt{N}}\boldsymbol{W}^\ell\right)\left(\frac{1}{\sqrt{D}}\boldsymbol{W}^0\right)\boldsymbol{x}. \tag{22}$$

In this section, we consider weights are updating these weights with gradient descent on the empirical (training) loss

$$\hat{\mathcal{L}}(t) = \frac{1}{P}\sum_{\mu=1}^{P}[\Delta_\mu(t)]^2 \ , \ \Delta_\mu(t) \equiv y_\mu - f(\boldsymbol{x}_\mu) = \frac{1}{\sqrt{D}}\boldsymbol{w}^\star \cdot \boldsymbol{x}_\mu + \sigma\epsilon_\mu - f(\boldsymbol{x}_\mu). \tag{23}$$

The update steps are performed using a raw learning rate of $N\eta\gamma_0^2$ following Bordelon & Pehlevan (2022), which gives

$$\boldsymbol{w}^L(t+1) = \boldsymbol{w}^L(t) - N\eta\gamma_0^2\nabla_{\boldsymbol{w}^L}\hat{\mathcal{L}}(t) \ , \ \boldsymbol{W}^\ell(t+1) = \boldsymbol{W}^\ell(t) - N\eta\gamma_0^2\nabla_{\boldsymbol{W}^\ell}\hat{\mathcal{L}}(t) \ , \ \ell \in \{0, ..., L-1\}. \tag{24}$$

Using the chain rule at each step $t$, we can decompose the weights at time $t$ into their initial values and all of their updates

$$\boldsymbol{W}^0(t) = \boldsymbol{W}^0(0) + \frac{\eta\gamma_0}{\sqrt{D}}\sum_{t'<t}\boldsymbol{g}^1(t')\boldsymbol{h}^0(t')^\top$$

$$\boldsymbol{W}^\ell(t) = \boldsymbol{W}^\ell(0) + \frac{\eta\gamma_0}{\sqrt{N}}\sum_{t'<t}\boldsymbol{g}^{\ell+1}(t')\boldsymbol{h}^\ell(t')^\top$$

$$\boldsymbol{w}^L(t) = \boldsymbol{w}^L(0) + \eta\gamma_0\sum_{t'<t}\boldsymbol{h}^L(t') \tag{25}$$

where the vectors $\boldsymbol{g}^\ell(t), \boldsymbol{h}^\ell(t) \in \mathbb{R}^N$ that we introduced defined in the following way

$$\boldsymbol{h}^0(t) = \frac{\sqrt{D}}{P}\boldsymbol{X}^\top\boldsymbol{\Delta}(t) \quad , \quad \boldsymbol{\Delta}(t) = \frac{1}{\sqrt{D}}\boldsymbol{X}\boldsymbol{v}(t) + \sigma\boldsymbol{\epsilon} \quad , \quad \boldsymbol{v}(t) = \boldsymbol{w}_\star - \frac{\sqrt{D}}{N\gamma_0}\boldsymbol{W}^0(t)^\top\boldsymbol{g}^1(t)$$

$$\boldsymbol{h}^{\ell+1}(t) = \frac{1}{\sqrt{N}}\boldsymbol{W}^\ell(t)\boldsymbol{h}^\ell(t) \ , \ \ell \in \{0, 1, ..., L-1\}$$

$$\boldsymbol{g}^L(t) = \boldsymbol{w}^L(t)$$

$$\boldsymbol{g}^\ell(t) = \frac{1}{\sqrt{N}}\boldsymbol{W}^\ell(t)^\top\boldsymbol{g}^{\ell+1}(t) \ , \ \ell \in \{1, ..., L-1\} \tag{26}$$

In the above definitions, we have introduced the data matrix $\boldsymbol{X} \in \mathbb{R}^{P\times D}$ and the noise in the target values $\boldsymbol{\epsilon} \in \mathbb{R}^D$. We can now isolate the dependence of these vectors on the random initial conditions $\boldsymbol{W}^\ell(0)$ and the updates from gradient descent. Expanding out the weight dynamics, we can close the $\boldsymbol{v}(t), \{\boldsymbol{h}^\ell(t), \boldsymbol{g}^\ell(t)\}_{\ell\in[L]}$ dynamics as

$$\boldsymbol{h}^{\ell+1}(t) = \boldsymbol{\chi}^{\ell+1}(t) + \eta\gamma_0\sum_{t'<t}C_h^\ell(t,t')\boldsymbol{g}^{\ell+1}(t')$$

$$\boldsymbol{g}^\ell(t) = \boldsymbol{\xi}^\ell(t) + \eta\gamma_0\sum_{t'<t}C_g^{\ell+1}(t,t')\boldsymbol{h}^\ell(t')$$

$$\boldsymbol{v}(t) = \boldsymbol{w}_\star - \frac{\sqrt{D}}{N\gamma_0}\boldsymbol{W}^0(0)^\top\boldsymbol{g}^1(t) - \eta\sum_{t'<t}C_g^1(t,t')\boldsymbol{h}^0(t')$$

where we introduced the following correlation functions

$$C_h^\ell(t, t') \equiv \frac{1}{N} \boldsymbol{h}^\ell(t) \cdot \boldsymbol{h}^\ell(t') \,, \; C_g^\ell(t, t') \equiv \frac{1}{N} \boldsymbol{g}^\ell(t) \cdot \boldsymbol{g}^\ell(t') \tag{27}$$

and the following fields that depend on the initial conditions $\boldsymbol{W}^\ell(0)$

$$\boldsymbol{\xi}^0(t) = \frac{\sqrt{D}}{N \gamma_0} \boldsymbol{W}^0(0)^\top \boldsymbol{g}^1(t) \,, \; \boldsymbol{\chi}^1(t) = \frac{1}{\sqrt{D}} \boldsymbol{W}^0(0) \boldsymbol{h}^0(t)$$

$$\boldsymbol{\chi}^{\ell+1}(t) = \frac{1}{\sqrt{N}} \boldsymbol{W}^\ell(0) \boldsymbol{h}^\ell(t) \,, \; \boldsymbol{\xi}^\ell(t) = \frac{1}{\sqrt{N}} \boldsymbol{W}^\ell(0)^\top \boldsymbol{g}^{\ell+1}(t). \tag{28}$$

These variables isolate the dependence on all of the random initial conditions for the weights. Combined with the defining equations for the variables $\boldsymbol{\Delta}(t), \boldsymbol{v}(t), \boldsymbol{h}^0(t)$, we will now aim to characterize the typical case dynamics over draws of the disorder $\mathcal{D}$

$$\mathcal{D} = \{\boldsymbol{X}, \boldsymbol{\epsilon}, \boldsymbol{W}^0(0), \boldsymbol{W}^1(0), ..., \boldsymbol{w}^L(0)\}. \tag{29}$$

Our goal is to characterize the typical case dynamics of the system over random draws of $\mathcal{D}$.

### A.1.2. COMPUTING THE DYNAMICAL DISTRIBUTION OVER DRAWS OF THE DISORDER

**Moment Generating Function** We now aim to characterize the joint distribution of $\boldsymbol{v}(t), \boldsymbol{\Delta}(t), \boldsymbol{h}^0(t), \{\boldsymbol{h}^\ell(t), \boldsymbol{g}^\ell(t)\}_{\ell=1}^L$ over draws of the random disorder $\mathcal{D}$. To do so, we will start by calculating the moment generating function of these fields with a Martin-Siggia-Rose integral (Martin et al., 1973) over the trajectories

$$Z[\boldsymbol{j}_v, \boldsymbol{j}_\Delta, \{\boldsymbol{j}_{h^\ell}, \boldsymbol{j}_{g^\ell}\}_{\ell \in [L]}] = \left\langle \exp \left( \sum_t \left[ \boldsymbol{j}_v(t) \cdot \boldsymbol{v}(t) + \boldsymbol{j}_\Delta(t) \cdot \boldsymbol{\Delta}(t) + \sum_\ell \boldsymbol{j}_{h^\ell}(t) \cdot \boldsymbol{h}^\ell(t) + \boldsymbol{j}_{g^\ell}(t) \cdot \boldsymbol{g}^\ell(t) \right] \right) \right\rangle_\mathcal{D}. \tag{30}$$

This object, once computed, enables computation of arbitrary moments from derivatives with respect to the $\boldsymbol{j}$ variables at $\boldsymbol{j} = 0$. For example, the two-point correlation between the $\boldsymbol{v}(t)$ variables has the form

$$\langle v_i(t) v_k(t') \rangle_\mathcal{D} = \frac{\partial^2}{\partial j_{v,i}(t) \partial j_{v,k}(t')} Z[\boldsymbol{j}_v, \boldsymbol{j}_\Delta, \{\boldsymbol{j}_{h^\ell}, \boldsymbol{j}_{g^\ell}\}_{\ell \in [L]}]. \tag{31}$$

**Why No Replicas?** At this point, we stress that the dynamics for our variables $\boldsymbol{v}(t), \boldsymbol{\Delta}(t), \boldsymbol{h}^\ell(t), \boldsymbol{g}^\ell(t)$ provide us a *complete and normalized joint distribution* unlike in static (replica) mean field calculations where the distribution of the variables of interest are only known up to a normalization constant (eg, where one must perform a quenched average of an observable $\mathcal{O}(\boldsymbol{\theta})$ for a Hamiltonian $\mathcal{H}(\boldsymbol{\theta}, \mathcal{D})$ which depends on disorder $\mathcal{D}$: $\left\langle \frac{\int d\boldsymbol{\theta} \exp(-\beta \mathcal{H}(\boldsymbol{\theta}, \mathcal{D})) \mathcal{O}(\boldsymbol{\theta})}{\int d\boldsymbol{\theta} \exp(-\beta \mathcal{H}(\boldsymbol{\theta}, \mathcal{D}))} \right\rangle_\mathcal{D}$ where we see the disorder appears in both numerator and denominator). This form of average is usually tackled with the replica method. In our case, this is not necessary since $Z$ directly captures the distribution of our trajectories over draws of disorder $\mathcal{D}$.

**Performing Averages** We now aim to compute $Z$ by averaging the dynamical trajectory over the draws of $\mathcal{D}$. To make this concrete, we represent the transition probabilities (conditional on $\mathcal{D}$) with Dirac-delta functions that encode the dynamics. We then represent the Dirac Delta function with a Fourier integral for each timestep $t$ as

$$1 = \int d\boldsymbol{\Delta}(t) \delta \left( \boldsymbol{\Delta}(t) - \frac{1}{\sqrt{D}} \boldsymbol{X} \boldsymbol{v}(t) - \sigma \boldsymbol{\epsilon} \right) = \int \frac{d\boldsymbol{\Delta}(t) d\hat{\boldsymbol{\Delta}}(t)}{(2\pi)^P} \exp \left( i\hat{\boldsymbol{\Delta}}(t) \cdot \left[ \boldsymbol{\Delta}(t) - \frac{1}{\sqrt{D}} \boldsymbol{X} \boldsymbol{v}(t) - \sigma \boldsymbol{\epsilon} \right] \right)$$

$$1 = \int d\boldsymbol{h}^0(t) \delta \left( \boldsymbol{h}^0(t) - \frac{\sqrt{D}}{P} \boldsymbol{X}^\top \boldsymbol{\Delta}(t) \right) = \int \frac{d\boldsymbol{h}^0(t) d\hat{\boldsymbol{h}}^0(t)}{(2\pi)^D} \exp \left( i\hat{\boldsymbol{h}}^0(t) \cdot \left[ \boldsymbol{h}^0(t) - \frac{\sqrt{D}}{P} \boldsymbol{X}^\top \boldsymbol{\Delta}(t) \right] \right) \tag{32}$$

...

Performing this for all of the relevant fields, we can begin to take averages over the terms involving each of the random matrices in the set of disorder $\mathcal{D}$.

**Data Average**  The average over the data matrix $\boldsymbol{X}$ involves terms of the form

$$\left\langle \exp\left( -i\,\mathrm{Tr}\boldsymbol{X}^\top \sum_{t=0}^\infty \left[ \frac{1}{\sqrt{D}}\hat{\boldsymbol{\Delta}}(t)\boldsymbol{v}(t)^\top + \frac{\sqrt{D}}{P}\boldsymbol{\Delta}(t)\hat{\boldsymbol{h}}^0(t)^\top \right] \right) \right\rangle_{\boldsymbol{X}} \tag{33}$$

$$= \exp\left( -\frac{1}{2}\sum_{tt'} \hat{\boldsymbol{\Delta}}(t)\cdot\hat{\boldsymbol{\Delta}}(t') \underbrace{\frac{1}{D}\boldsymbol{v}(t)\cdot\boldsymbol{v}(t')}_{C_v(t,t')} - \frac{D}{2P}\sum_{tt'} \hat{\boldsymbol{h}}^0(t)\cdot\hat{\boldsymbol{h}}^0(t') \underbrace{\frac{1}{P}\boldsymbol{\Delta}(t)\cdot\boldsymbol{\Delta}(t')}_{C_\Delta(t,t')} \right) \tag{34}$$

$$\times \exp\left( -\frac{1}{P}\sum_{tt'} \underbrace{\hat{\boldsymbol{\Delta}}(t)\cdot\boldsymbol{\Delta}(t)}_{iPR_\Delta(t',t)} \underbrace{\boldsymbol{v}(t)\cdot\hat{\boldsymbol{h}}^0(t')}_{iDR^0_{vu}(t,t')} \right). \tag{35}$$

The resulting underlined terms reveal the following correlation and response functions induced by averaging over this fixed data matrix

$$C_v(t,t') \equiv \frac{1}{D}\boldsymbol{v}(t)\cdot\boldsymbol{v}(t') \,,\;\; C_\Delta(t,t') \equiv \frac{1}{P}\boldsymbol{\Delta}(t)\cdot\boldsymbol{\Delta}(t')$$

$$R^0_{vu}(t,t') \equiv -\frac{i}{D}\boldsymbol{v}(t)\cdot\hat{\boldsymbol{h}}^0(t') \,,\;\; R_\Delta(t,t') \equiv -\frac{i}{P}\boldsymbol{\Delta}(t)\cdot\hat{\boldsymbol{\Delta}}(t') \tag{36}$$

These correlation and response functions will be a subset of the order parameters of the mean field theory theory (more will arise from other disorder averages). We now introduce delta functions to enforce the definitions of these order parameters

$$1 = \int \frac{dC_v(t,t')d\hat{C}_v(t,t')}{4\pi D^{-1}} \exp\left( \frac{D}{2}\hat{C}_v(t,t')C_v(t,t') - \frac{1}{2}\hat{C}_v(t,t')\boldsymbol{v}(t)\cdot\boldsymbol{v}(t') \right) \tag{37}$$

$$1 = \int \frac{dC_\Delta(t,t')d\hat{C}_\Delta(t,t')}{4\pi P^{-1}} \exp\left( \frac{P}{2}\hat{C}_\Delta(t,t')C_\Delta(t,t') - \frac{1}{2}\hat{C}_\Delta(t,t')\boldsymbol{\Delta}(t)\cdot\boldsymbol{\Delta}(t') \right) \tag{38}$$

For the response function, the conjugate response function to $R^0_{vu}$ is $R_\Delta$ and vice versa so it suffices to just enforce the definition for one of them. For concreteness, we can enforce the definition of $R^0_{vu}(t,t')$

$$1 = \int \frac{dR^0_{vu}(t,t')dR_\Delta(t',t)}{2\pi D^{-1}} \exp\left( -DR^0_{vu}(t,t')R_\Delta(t',t) - i\boldsymbol{v}(t)\cdot\hat{\boldsymbol{h}}^0(t')R_\Delta(t',t) \right) \tag{39}$$

**First Layer Weight Matrix**  We now examine the behavior of the first weight matrix $\boldsymbol{W}^0(0) \in \mathbb{R}^{N\times D}$ which is rectangular. For this matrix the two relevant dynamical fields are

$$\boldsymbol{\chi}^1(t) = \frac{1}{\sqrt{D}}\boldsymbol{W}^0(0)\boldsymbol{h}^0(t) \,,\;\; \boldsymbol{\xi}^0(t) = \frac{\sqrt{D}}{N\gamma_0}\boldsymbol{W}^0(t)\boldsymbol{g}^1(t) \tag{40}$$

Again, we introduce Dirac delta functions for these fields and need to average

$$\left\langle \exp\left( -i\,\mathrm{Tr}\boldsymbol{W}^0(0)^\top \sum_{t=0}^\infty \left[ \frac{1}{\sqrt{D}}\hat{\boldsymbol{\chi}}^1(t)\boldsymbol{h}^0(t)^\top + \frac{\sqrt{D}}{N\gamma_0}\boldsymbol{g}^1(t)\hat{\boldsymbol{\xi}}^0(t)^\top \right] \right) \right\rangle_{\boldsymbol{W}^0(0)} \tag{41}$$

$$= \exp\left( -\frac{1}{2}\sum_{tt'} \hat{\boldsymbol{\chi}}^1(t)\cdot\hat{\boldsymbol{\chi}}^1(t') \underbrace{\frac{1}{D}\boldsymbol{h}^0(t)\cdot\boldsymbol{h}^0(t')}_{C^0_h(t,t')} - \frac{D}{2N\gamma_0^2}\sum_{tt'} \hat{\boldsymbol{\xi}}^0(t)\cdot\hat{\boldsymbol{\xi}}^0(t') \underbrace{\frac{1}{N}\boldsymbol{g}^1(t)\cdot\boldsymbol{g}^1(t')}_{C^1_g(t,t')} \right) \tag{42}$$

$$\times \exp\left( -\frac{1}{N\gamma_0}\sum_{tt'} \underbrace{\hat{\boldsymbol{\chi}}^1(t)\cdot\boldsymbol{g}^1(t')}_{iNR^1_{gu}(t',t)} \underbrace{\boldsymbol{h}^0(t)\cdot\hat{\boldsymbol{\xi}}^0(t')}_{iDR^0_{hr}(t,t')} \right) \tag{43}$$

where we introduced the order parameters

$$C_h^0(t,t') \equiv \frac{1}{D} \boldsymbol{h}^0(t) \cdot \boldsymbol{h}^0(t') \,, \; C_g^1(t,t') \equiv \frac{1}{N} \boldsymbol{g}^1(t) \cdot \boldsymbol{g}^1(t') \tag{44}$$

$$R_{gu}^1(t,t') \equiv \frac{1}{N} \boldsymbol{g}^1(t) \cdot \hat{\boldsymbol{\chi}}^1(t') \,, \; R_{hr}^0(t,t') \equiv \frac{1}{D} \boldsymbol{h}^0(t) \cdot \hat{\boldsymbol{\xi}}^0(t') \tag{45}$$

As before, we introduce Dirac delta functions to enforce the definitions of these variables.

**Hidden Layers** Lastly, we examine the disorder averages for the hidden layers $\boldsymbol{W}^\ell(0) \in \mathbb{R}^{N \times N}$ which are square matrices. The random fields that depend on the random matrix $\boldsymbol{W}^\ell(0)$ are

$$\boldsymbol{\chi}^{\ell+1}(t) = \frac{1}{\sqrt{N}} \boldsymbol{W}^\ell(0) \boldsymbol{h}^\ell(t) \,, \; \boldsymbol{\xi}^\ell(t) = \frac{1}{\sqrt{N}} \boldsymbol{W}^\ell(0)^\top \boldsymbol{g}^{\ell+1}(t). \tag{46}$$

Averaging their trajectories over the disorder $\boldsymbol{W}^\ell(0)$ gives us

$$\left\langle \exp\left( \frac{i}{\sqrt{N}} \mathrm{Tr} \boldsymbol{W}^\ell(0)^\top \sum_t \left[ \hat{\boldsymbol{\chi}}^{\ell+1}(t) \boldsymbol{h}^\ell(t)^\top + \boldsymbol{g}^{\ell+1}(t) \hat{\boldsymbol{\xi}}^\ell(t)^\top \right] \right) \right\rangle_{\boldsymbol{W}^\ell(0)}$$

$$= \exp\left( -\frac{1}{2} \sum_{tt'} \hat{\boldsymbol{\chi}}^{\ell+1}(t) \cdot \hat{\boldsymbol{\chi}}^{\ell+1}(t') \underbrace{\frac{1}{N} \boldsymbol{h}^\ell(t) \cdot \boldsymbol{h}^\ell(t')}_{C_h^\ell(t,t')} - \frac{1}{2} \sum_{tt'} \hat{\boldsymbol{\xi}}^\ell(t) \cdot \hat{\boldsymbol{\xi}}^\ell(t') \underbrace{\frac{1}{N} \boldsymbol{g}^{\ell+1}(t) \cdot \boldsymbol{g}^{\ell+1}(t')}_{C_g^{\ell+1}(t,t')} \right)$$

$$\times \exp\left( -\frac{1}{N} \sum_{tt'} \underbrace{\hat{\boldsymbol{\chi}}^{\ell+1}(t) \cdot \boldsymbol{g}^{\ell+1}(t')}_{iNR_{gu}^{\ell+1}(t',t)} \underbrace{\boldsymbol{h}^\ell(t) \cdot \hat{\boldsymbol{\xi}}^\ell(t')}_{iNR_{hr}^\ell(t,t')} \right) \tag{47}$$

where the relevant order parameters are

$$C_h^\ell(t,t') \equiv \frac{1}{N} \boldsymbol{h}^\ell(t) \cdot \boldsymbol{h}^\ell(t') \,, \; C_g^{\ell+1}(t,t') \equiv \frac{1}{N} \boldsymbol{g}^{\ell+1}(t) \cdot \boldsymbol{g}^{\ell+1}(t')$$

$$R_{hr}^\ell(t,t') \equiv -\frac{i}{N} \boldsymbol{h}^\ell(t) \cdot \hat{\boldsymbol{\xi}}^\ell(t') \,, \; R_{gu}^{\ell+1}(t,t') \equiv -\frac{i}{N} \boldsymbol{g}^{\ell+1}(t) \cdot \hat{\boldsymbol{\chi}}^{\ell+1}(t') \tag{48}$$

As before, we introduce these order parameters with dual Fourier variables.

**Proportional Scaling Regime** We notice that many of the terms that arise in the integrals depend on ratios such as $\frac{N}{D}$ and $\frac{P}{D}$. To deal with these terms we introduce definitions for these ratios

$$\alpha \equiv \frac{P}{D} \,, \; \nu \equiv \frac{N}{D}. \tag{49}$$

One can interpret, at fixed $D$, as $\alpha$ and $\nu$ representing an increase in the quantity of data or the width of the network respectively. With these constants $\alpha, \nu$ introduced, we can now consider how the moment generating function $Z$ behaves as $D \to \infty$. However, before this, we must organize and simplify the remaining integrals to enable taking the $D \to \infty$ limit.

**Decoupling The Single Site Integrals** Now we aim to express the integrals over the variables $\boldsymbol{\Delta}(t), \boldsymbol{v}(t), \boldsymbol{\chi}^\ell(t), \boldsymbol{\xi}^\ell(t)$ and their conjugates

$$\int \prod_{t=0}^\infty \frac{d\boldsymbol{\Delta}(t) d\hat{\boldsymbol{\Delta}}(t)}{(2\pi)^P} \exp\left( -\frac{1}{2} \sum_{tt'} \hat{\boldsymbol{\Delta}}(t) \cdot \hat{\boldsymbol{\Delta}}(t') C_v(t,t') + i \sum_{tt'} \hat{\boldsymbol{\Delta}}(t) \cdot \left( \boldsymbol{\Delta}(t') \delta(t-t') - R_{vu}^0(t,t') \boldsymbol{\Delta}(t') \right) \right)$$

$$\times \exp\left( \sum_t \boldsymbol{j}_\Delta(t) \cdot \boldsymbol{\Delta}(t) - \frac{1}{2} \sum_{tt'} \hat{C}_\Delta(t,t') \boldsymbol{\Delta}(t) \cdot \boldsymbol{\Delta}(t') \right)$$

$$= \prod_{\mu=1}^P \mathcal{Z}_\Delta[C_v, \hat{C}_\Delta, R_{vu}^0, j_{\Delta,\mu}(t)] \tag{50}$$

where we introduced the *single site moment generating function* for the $\Delta(t)$ fields that has the form of an integral over a single entry (scalar stochastic process) of the original $\boldsymbol{\Delta}(t)$ vector

$$Z_\Delta = \int \prod_{t=0}^\infty \frac{d\Delta(t)d\hat{\Delta}(t)}{2\pi} \exp\left(-\frac{1}{2}\sum_{tt'} \hat{\Delta}(t)\hat{\Delta}(t')C_v(t,t') + i\sum_{tt'} \hat{\Delta}(t)\left(\Delta(t')\delta(t-t') - R_{vu}^0(t,t')\Delta(t')\right)\right)$$

$$\times \exp\left(\sum_t j_\Delta(t)\Delta(t) - \frac{1}{2}\sum_{tt'}\hat{C}_\Delta(t,t')\Delta(t)\Delta(t')\right) \tag{51}$$

From our disorder average, we also see that there is statistical coupling between $\boldsymbol{v}(t), \boldsymbol{\xi}^0(t)$ and $\boldsymbol{h}^0(t)$ variables. We let the joint moment generating function for this set of variables be $\mathcal{Z}_0$ (for layer $\ell = 0$), and express it as

$$\mathcal{Z}_0 = \int \prod_{t=0}^\infty \frac{dv(t)d\xi^0(t)d\hat{\xi}^0(t)dh^0(t)d\hat{h}^0(t)}{4\pi^2}$$

$$\exp\left(-\frac{1}{2\alpha}\sum_{tt'}\hat{h}^0(t)\hat{h}^0(t')C_\Delta(t,t') - \frac{1}{2\gamma_0^2\nu}\sum_{tt'}\hat{\xi}^0(t)\hat{\xi}^0(t')C_g^1(t,t')\right)$$

$$\exp\left(i\sum_{tt'}\hat{h}^0(t)\left(\delta(t-t')h^0(t') - R_\Delta(t,t')v(t')\right) + \sum_t j_{h^0}(t)h^0(t)\right)$$

$$\exp\left(i\sum_{tt'}\hat{\xi}^0(t)\left(\delta(t-t')\xi^0(t') - \frac{1}{\gamma_0}R_{gu}^1(t,t')h^0(t')\right) + \sum_t j_v(t)v(t)\right)$$

$$\prod_t \delta\left(v(t) - w^\star + \xi^0(t) + \eta\sum_{t'<t}C_g^1(t,t')h^0(t')\right) \tag{52}$$

For the first hidden layer ($\ell = 1$), we have the following joint single site MGF

$$\mathcal{Z}_1 = \int \prod_t \frac{d\chi^1(t)d\hat{\chi}^1(t)d\xi^1(t)d\hat{\xi}^1(t)dh^1(t)dg^1(t)}{4\pi^2}$$

$$\exp\left(-\frac{1}{2}\sum_{tt'}\hat{\chi}^1(t)\hat{\chi}^1(t')C_h^0(t,t') - \frac{1}{2}\sum_{tt'}\hat{\xi}^1(t)\hat{\xi}^1(t')C_g^2(t,t')\right)$$

$$\exp\left(i\sum_{tt'}\hat{\chi}^1(t)\left(\delta(t-t')\chi^1(t') - \frac{1}{\nu\gamma_0}R_{hr}^0(t,t')g^1(t')\right)\right)$$

$$\exp\left(i\sum_{tt'}\hat{\xi}^1(t)\left(\delta(t-t')\xi^1(t') - R_{gu}^2(t,t')h^1(t')\right)\right)$$

$$\exp\left(\sum_t \left[j_{h^1}(t)h^1(t) + j_{g^1}(t)g^1(t)\right]\right)$$

$$\prod_t \delta\left(h^1(t) - \chi^1(t) - \eta\gamma_0\sum_{t'<t}C_h^0(t,t')g^1(t')\right)\delta\left(g^1(t) - \xi^1(t) - \eta\gamma_0\sum_{t'<t}C_g^2(t,t')h^1(t')\right) \tag{53}$$

For the hidden layers $\ell \in \{2, ..., L-1\}$ we have the following single site MGFs

$$
\mathcal{Z}_\ell = \int \prod_t \frac{d\chi^\ell(t) d\hat{\chi}^\ell(t) d\xi^\ell(t) d\hat{\xi}^\ell(t) dh^\ell(t) dg^\ell(t)}{4\pi^2}
$$

$$
\exp\left( -\frac{1}{2} \sum_{tt'} \hat{\chi}^\ell(t) \hat{\chi}^\ell(t') C_h^{\ell-1}(t,t') - \frac{1}{2} \sum_{tt'} \hat{\xi}^\ell(t) \hat{\xi}^\ell(t') C_g^{\ell+1}(t,t') \right)
$$

$$
\exp\left( i \sum_{tt'} \hat{\chi}^\ell(t) \left( \delta(t-t')\chi^\ell(t') - R_{hr}^{\ell-1}(t,t') g^\ell(t') \right) \right)
$$

$$
\exp\left( i \sum_{tt'} \hat{\xi}^\ell(t) \left( \delta(t-t')\xi^\ell(t') - R_{gu}^{\ell+1}(t,t') h^\ell(t') \right) \right)
$$

$$
\exp\left( \sum_t \left[ j_{h^\ell}(t) h^\ell(t) + j_{g^\ell}(t) g^\ell(t) \right] \right)
$$

$$
\prod_t \delta\left( h^\ell(t) - \chi^\ell(t) - \eta\gamma_0 \sum_{t'<t} C_h^{\ell-1}(t,t') g^\ell(t') \right) \delta\left( g^\ell(t) - \xi^\ell(t) - \eta\gamma_0 \sum_{t'<t} C_g^{\ell+1}(t,t') h^1(t') \right) \quad (54)
$$

The final layer $\ell = L$ has the following single site MGF

$$
\mathcal{Z}_L = \int \frac{dh^L(t) d\hat{\chi}^L(t) d\chi^L(t) dg^L(t)}{2\pi}
$$

$$
\exp\left( -\frac{1}{2} \sum_{tt'} \hat{\chi}^L(t) \hat{\chi}^L(t') C_h^{L-1}(t,t') + i \sum_{tt'} \hat{\chi}^L(t) \left( \chi^L(t')\delta(t-t') - R_{hr}^{L-1}(t,t') g^L(t') \right) \right)
$$

$$
\prod_t \delta\left( h^L(t) - \chi^L(t) - \eta\gamma_0 \sum_{t'<t} C_h^{L-1}(t,t') g^L(t') \right)
$$

$$
\left\langle \prod_t \delta\left( g^L(t) - g^L(0) - \eta\gamma_0 \sum_{t'<t} h^L(t') \right) \right\rangle_{g^L(0) \sim \mathcal{N}(0,1)}
$$

$$
\exp\left( \sum_t \left[ j_{h^L}(t) h^L(t) + j_{g^L}(t) g^L(t) \right] \right) \quad (55)
$$

Now we have reduced all of the field path integrals into single site moment generating functions.

**DMFT Action**   We now group all of the correlation and response functions, as well as their conjugate order parameters into a list $q$ and combine all of the source variables into a list $j$. The moment generating function can be written as

$$
Z[j] = \int dq \exp\left( -D\mathcal{S}[q,j] \right) \quad (56)
$$

where $\mathcal{S}$ is an $\mathcal{O}(1)$ DMFT action which has the form

$$
\mathcal{S} = -\frac{1}{2} \sum_{tt'} \left[ C_v(t,t')\hat{C}_v(t,t') + \alpha C_\Delta(t,t')\hat{C}_\Delta(t,t') \right] - \frac{\nu}{2} \sum_\ell \sum_{tt'} \left[ C_{h^\ell}(t,t')\hat{C}_{h^\ell}(t,t') + C_g^\ell(t,t')\hat{C}_g^\ell(t,t') \right]
$$

$$
+ \sum_{tt'} R_{vu}^0(t,t') R_\Delta(t',t) + \gamma_0^{-1} \sum_{tt'} R_{hr}^0(t,t') R_{gu}^1(t',t) + \nu \sum_\ell \sum_{tt'} R_{hr}^\ell(t,t') R_{gu}^{\ell+1}(t',t)
$$

$$
- \frac{1}{D} \sum_{\mu=1}^P \ln \mathcal{Z}_\Delta[j_{\Delta,\mu}] - \frac{1}{D} \sum_{k=1}^D \mathcal{Z}_0[j_{h^0,k}, j_{v,k}] - \frac{1}{D} \sum_{\ell=1}^L \sum_{i=1}^N \ln \mathcal{Z}_\ell[j_{h^\ell,i}, j_{g^\ell,i}] \quad (57)
$$

**Saddle Point Equations**  We are now in a position to take the $D \to \infty$ limit. In this limit, the integral over $\boldsymbol{q}$ is dominated by the saddle point of $\mathcal{S}$. The resulting saddle point equations $\frac{\partial \mathcal{S}}{\partial \boldsymbol{q}} = 0$. We start by differentiating with respect to the conjugate order parameters for the correlation functions which give

$$\frac{\partial \mathcal{S}}{\partial \hat{C}_v(t,t')} = -\frac{1}{2} C_v(t,t') + \frac{1}{2D} \sum_{k=1}^{D} \langle v(t)v(t') \rangle_k = 0$$

$$\frac{\partial \mathcal{S}}{\partial \hat{C}_h^0(t,t')} = -\frac{1}{2} C_h^0(t,t') + \frac{1}{2D} \sum_{k=1}^{D} \langle h^0(t)h^0(t') \rangle_k = 0$$

$$\frac{\partial \mathcal{S}}{\partial \hat{C}_\Delta(t,t')} = -\frac{\alpha}{2} C_\Delta(t,t') + \frac{1}{2D} \sum_{\mu=1}^{P} \langle \Delta(t)\Delta(t') \rangle_\mu = 0$$

$$\frac{\partial \mathcal{S}}{\partial \hat{C}_h^\ell(t,t')} = -\frac{\nu}{2} C_h^\ell(t,t') + \frac{1}{2D} \sum_{i=1}^{N} \langle h^\ell(t)h^\ell(t') \rangle_i = 0$$

$$\frac{\partial \mathcal{S}}{\partial \hat{C}_g^\ell(t,t')} = -\frac{\nu}{2} C_h^\ell(t,t') + \frac{1}{2D} \sum_{i=1}^{N} \langle g^\ell(t)g^\ell(t') \rangle_i = 0 \tag{58}$$

where $\langle \rangle_\mu$ represents an average over the single site distribution defined by the moment generating function $\mathcal{Z}_\Delta[j_{\Delta,\mu}]$. Similarly $\langle \rangle_k, \langle \rangle_i$ denote averages over the distributions defined by the single site moment generating functions $\mathcal{Z}_0[j_k], \{\mathcal{Z}_\ell[j_i]\}_{\ell=1}^L$. We note that in the limit of zero source $\boldsymbol{j} \to 0$ these averages all become identical (we will examine this zero source limit shortly). Next, we can work out the saddle point equations for the response functions

$$\frac{\partial \mathcal{S}}{\partial R_{vu}^0(t',t)} = R_\Delta(t,t') + \frac{i}{P} \sum_{\mu=1}^{P} \left\langle \Delta(t)\hat{\Delta}(t') \right\rangle_\mu = 0$$

$$\frac{\partial \mathcal{S}}{\partial R_\Delta(t',t)} = R_{vu}(t,t') + \frac{i}{D} \sum_{k=1}^{D} \left\langle v(t)\hat{h}^0(t') \right\rangle_k = 0$$

$$\frac{\partial \mathcal{S}}{\partial R_{hr}^0(t',t)} = \gamma_0^{-1} R_{gu}^1(t,t') + \frac{i}{N\gamma_0} \sum_{i=1}^{N} \left\langle g^1(t)\hat{\xi}^1(t') \right\rangle_i = 0$$

$$\frac{\partial \mathcal{S}}{\partial R_{gu}^1(t',t)} = \gamma_0^{-1} R_{hr}^0(t,t') + \frac{i}{D\gamma_0} \sum_{k=1}^{D} \langle h^0(t)\hat{\chi}^0(t') \rangle_k = 0$$

$$\frac{\partial \mathcal{S}}{\partial R_{hr}^\ell(t',t)} = \nu R_{gu}^{\ell+1}(t,t') + \frac{i}{D} \sum_{i=1}^{N} \left\langle g^{\ell+1}(t)\hat{\xi}^{\ell+1}(t') \right\rangle_i = 0$$

$$\frac{\partial \mathcal{S}}{\partial R_{gu}^{\ell+1}(t',t)} = \nu R_{hr}^\ell(t,t') + \frac{i}{D} \sum_{i=1}^{N} \langle h^\ell(t)\hat{\chi}^\ell(t') \rangle_i = 0 \tag{59}$$

Lastly, we have a collection of saddle point equations $\frac{\partial \mathcal{S}}{\partial C} = 0$ that define the conjugate order parameters $\hat{C} = 0$ which must vanish at the saddle point (Crisanti & Sompolinsky, 2018; Mignacco et al., 2020; Bordelon & Pehlevan, 2022). Thus we have the following remaining set of equations

$$\hat{C}_\Delta(t,t') = 0 \,, \ \hat{C}_v(t,t') = 0 \,, \ \hat{C}_h^\ell(t,t') = 0 \,, \ \hat{C}_g^\ell(t,t') = 0. \tag{60}$$

**Hubbard Transformation**  Now that we have argued that the correlation and response functions take on deterministic values in the $D \to \infty$ limit, we can represent the quadratic terms in log-density in $\hat{\Delta}, \hat{\chi}, \hat{\xi}$ fields as linear averages Gaussian

random variables $u_\Delta(t), u^\ell(t), r^\ell(t)$, giving

$$
\exp\left(-\frac{1}{2}\sum_{tt'}\hat{\Delta}(t)\hat{\Delta}(t')C_v(t,t')\right) = \left\langle \exp\left(-i\sum_t \hat{\Delta}(t)u_\Delta(t)\right)\right\rangle_{u_\Delta \sim \mathcal{N}(0,C_v)}
$$

$$
\exp\left(-\frac{1}{2\alpha}\sum_{tt'}\hat{h}^0(t)\hat{h}^0(t')C_\Delta(t,t')\right) = \left\langle \exp\left(-i\sum_t \hat{h}^0(t)u^0(t)\right)\right\rangle_{u^0 \sim \mathcal{N}(0,\alpha^{-1}C_\Delta)}
$$

$$
\exp\left(-\frac{1}{2}\sum_{tt'}\hat{\chi}^\ell(t)\hat{\chi}^\ell(t')C_h^{\ell-1}(t,t')\right) = \left\langle \exp\left(-i\sum_t \hat{\chi}^\ell(t)u^\ell(t)\right)\right\rangle_{u^\ell \sim \mathcal{N}\left(0,C_h^{\ell-1}\right)}
$$

$$
\exp\left(-\frac{1}{2\gamma_0^2\nu}\sum_{tt'}\hat{\xi}^0(t)\hat{\xi}^0(t')C_g^1(t,t')\right) = \left\langle \exp\left(-i\sum_t \hat{\xi}^0(t)r^0(t)\right)\right\rangle_{r^0 \sim \mathcal{N}\left(0,\gamma_0^{-2}\nu^{-1}C_h^{\ell-1}\right)}
$$

$$
\exp\left(-\frac{1}{2}\sum_{tt'}\hat{\xi}^\ell(t)\hat{\xi}^\ell(t')C_g^{\ell+1}(t,t')\right) = \left\langle \exp\left(-i\sum_t \hat{\xi}^\ell(t)r^\ell(t)\right)\right\rangle_{r^0 \sim \mathcal{N}\left(0,C_g^{\ell+1}\right)} \tag{61}
$$

After introducing these Gaussian random variables, we can now perform integrals over the Fourier variables $\hat{\Delta}, \hat{h}^0, \hat{\chi}^\ell, \hat{\xi}^\ell$ to obtain defining equations for the random variables of interest. For example, for the $\Delta(t)$ random variable,

$$
\int \prod_t \frac{d\hat{\Delta}(t)}{2\pi}\left\langle \exp\left(i\sum_t \hat{\Delta}(t)\left[\Delta(t) - u_\Delta(t) - \sum_{t'} R_{vu}^0(t,t')\Delta(t')\right]\right)\right\rangle_{u_\Delta}
$$

$$
= \left\langle \delta\left(\Delta(t) - u_\Delta(t) - \sum_{t'} R_{vu}^0(t,t')\Delta(t')\right)\right\rangle_{u_\Delta(t) \sim \mathcal{N}(0,C_v)}
$$

$$
\implies \Delta(t) = u_\Delta(t) + \sum_{t'} R_{vu}^0(t,t')\Delta(t') , \; u_\Delta(t) \sim \mathcal{N}(0,C_v) \tag{62}
$$

We can repeat this for all of the remaining fields which give defining stochastic processes for them. The general result is that the random fields are driven by a noise component (like $u_\Delta(t)$ above) as well as a memory term involving a response function (like $R_{vu}^0(t,t')$ above).

**Simplifying the Response Functions**  We note that the response functions involve averages over the $\hat{\Delta}, \hat{\chi}^\ell, \hat{\xi}^\ell$ variables, which we now argue can be viewed as derivatives with respect to the Hubbard variable. For simplicity, we will now examine the zero source limit where all averages are identical.

$$
R_\Delta(t,t') = -i\left\langle \Delta(t)\hat{\Delta}(t')\right\rangle
$$

$$
= -i\int\left[\prod_t \frac{d\Delta(t)d\hat{\Delta}(t)}{2\pi}\right]\Delta(t)\hat{\Delta}(t')\left\langle \exp\left(i\sum_t \hat{\Delta}(t)\left[\Delta(t) - u_\Delta(t) - \sum_{t'} R_{vu}^0(t,t')\Delta(t')\right]\right)\right\rangle_{u_\Delta}
$$

$$
= \int\left[\prod_t \frac{d\Delta(t)d\hat{\Delta}(t)}{2\pi}\right]\Delta(t)\left\langle \frac{\partial}{\partial u_\Delta(t')}\exp\left(i\sum_t \hat{\Delta}(t)\left[\Delta(t) - u_\Delta(t) - \sum_{t'} R_{vu}^0(t,t')\Delta(t')\right]\right)\right\rangle_{u_\Delta}
$$

$$
= \sum_{t''}\left\langle \Delta(t)[C_v]^{-1}(t',t'')u_\Delta(t'')\right\rangle_{u_\Delta} = \left\langle \frac{\partial\Delta(t)}{\partial u_\Delta(t')}\right\rangle_{u_\Delta} \tag{63}
$$

which holds via integration by parts and Stein's lemma. Repeating this for each of the fields, we find the following response function definitions from the saddle point equations

$$
R_\Delta(t,t') = \left\langle \frac{\partial\Delta(t)}{\partial u_\Delta(t')}\right\rangle , \; R_{vu}^0(t,t') = \left\langle \frac{\partial v(t)}{\partial u^0(t')}\right\rangle , \; R_{hr}^0(t,t') = \left\langle \frac{\partial h^0(t)}{\partial r^0(t')}\right\rangle
$$

$$
R_{hr}^\ell(t,t') = \left\langle \frac{\partial h^\ell(t)}{\partial r^\ell(t')}\right\rangle , \; R_{gu}^\ell(t,t') = \left\langle \frac{\partial g^\ell(t)}{\partial u^\ell(t')}\right\rangle . \tag{64}
$$

### A.2. Final DMFT Equations

Combining the above equations at zero source $j \to 0$ after our Hubbard transformations, we arrive at the final single site equations for our fields. The DMFT equations that describe the single site stochastic processes are

$$v(t) = w^\star - r^0(t) - \sum_{t' < t} \left[ \frac{1}{\gamma_0} R_{gu}^1(t, t') + \eta \gamma_0 C_g^1(t, t') \right] h^0(t') \,, \; r^0(t) \sim \mathcal{GP}\left(0, \frac{1}{\nu \gamma_0^2} G^1\right)$$

$$\Delta(t) = u_\Delta(t) + \frac{1}{\alpha} \sum_{t' < t} R_{vu}^0(t, t') \Delta(t') + \sigma \epsilon \,, \; u_\Delta(t) \sim \mathcal{GP}(0, C_v) \,, \; \epsilon \sim \mathcal{N}(0, 1)$$

$$h^0(t) = u^0(t) + \sum_{t' < t} R_\Delta(t, t') v(t') \,, \; u^0(t) \sim \mathcal{GP}\left(0, \frac{1}{\alpha} C_\Delta\right)$$

$$h^1(t) = u^1(t) + \sum_{t' < t} \left[ \frac{1}{\nu \gamma_0} R_{hr}^0(t, t') + \eta \gamma_0 C_h^0(t, t') \right] g^1(t') \,, \; u^1(t) \sim \mathcal{GP}(0, C_h^0)$$

$$h^\ell(t) = u^\ell(t) + \sum_{t' < t} \left[ R_{hr}^{\ell-1}(t, t') + \eta \gamma_0 C_h^{\ell-1}(t, t') \right] g^\ell(t') \,, \; u^\ell(t) \sim \mathcal{GP}(0, C_h^{\ell-1})$$

$$g^\ell(t) = r^\ell(t) + \sum_{t' < t} \left[ R_{gu}^{\ell+1}(t, t') + \eta \gamma_0 C_g^{\ell+1}(t, t') \right] h^\ell(t') \,, \; r^\ell(t) \sim \mathcal{GP}(0, C_g^{\ell+1}) \tag{65}$$

From these equations, all necessary response functions can be computed by differentiating over the noise variables $\{u, r\}$.

$$R_\Delta(t, t') = \left\langle \frac{\partial \Delta(t)}{\partial u_\Delta(t')} \right\rangle \,, \; R_{vu}^0(t, t') = \left\langle \frac{\partial v(t)}{\partial u^0(t')} \right\rangle \,, \; R_{hr}^0(t, t') = \left\langle \frac{\partial h^0(t)}{\partial r^0(t')} \right\rangle$$

$$R_{hr}^\ell(t, t') = \left\langle \frac{\partial h^\ell(t)}{\partial r^\ell(t')} \right\rangle \,, \; R_{gu}^\ell(t, t') = \left\langle \frac{\partial g^\ell(t)}{\partial u^\ell(t')} \right\rangle \tag{66}$$

Similarly, the correlation functions can be computed as averages over these noise variables

$$C_v(t, t') = \langle v(t) v(t') \rangle \,, \; C_\Delta(t, t') = \langle \Delta(t) \Delta(t') \rangle$$

$$C_h^\ell(t, t') = \langle h^\ell(t) h^\ell(t') \rangle \,, \; C_g^\ell(t, t') = \langle g^\ell(t) g^\ell(t') \rangle \tag{67}$$

We note again that the test and train losses can be obtained from the time-time diagonal of the correlation functions.

$$\mathcal{L}(t) = C_v(t, t) + \sigma^2 \,, \; \hat{\mathcal{L}}(t) = C_\Delta(t, t) \tag{68}$$

### A.3. Centering: Subtracting the Initial Predictor for Lazy Limit Stability

At finite values of $\nu$ our theory will diverge if the lazy limit $\gamma_0 \to 0$ is taken. This is due to a divergence of the predictor variance $C_v(t, t)$ which can be eliminated with a *centering* operation. Under centering we define the vector $\boldsymbol{v}(t)$ as

$$\boldsymbol{v}(t) = \boldsymbol{w}^\star - \boldsymbol{\xi}^0(t) + \boldsymbol{\xi}^0(0) + \eta \sum_{t' < t} C_g^1(t, t') \boldsymbol{h}^0(t') \tag{69}$$

The key adjustment to the DMFT equations from this operation is

$$v(t) = w_\star - r^0(t) + r^0(0) - \frac{1}{\gamma_0} \sum_{t' < t} R_{gu}^1(t, t') h^0(t') - \eta \sum_{t' < t} C_g^1(t, t') h^0(t')$$

The correlation functions that need to be tracked are $\{C_v, C_h^0, C_h^1, ..., C_h^{L-1}, C_g^1, C_g^2, ..., C_g^L\}$ while we need the following response functions $\{R_\Delta, R_{hr}^0, ..., R_{hr}^{L-1}, R_{gu}^1, ..., R_{gu}^L\}$.

## B. Comparing Parameterizations

In this section, we briefly comment on how the maximal stable learning rate scales with $\nu$ in NTK and in mean field/$\mu$P parameterization. NTK parameterization is defined as scaling the network so that $\gamma_0 = \Theta\left(\frac{1}{\sqrt{\nu}}\right)$. However, a consequence of this choice is that the change in the correlation functions $C_g(t, t')$ under a fixed number of steps of optimization will decrease with $\gamma_0$ at fixed learning rate. As a consequence, higher learning rates can be tolerated at larger widths. However, once the network is wide enough to be sufficiently close to the kernel regime, the maximum stable learning rate stabilizes as the dynamics become asymptotically linear. A comparison showing this trend is provided in 8.

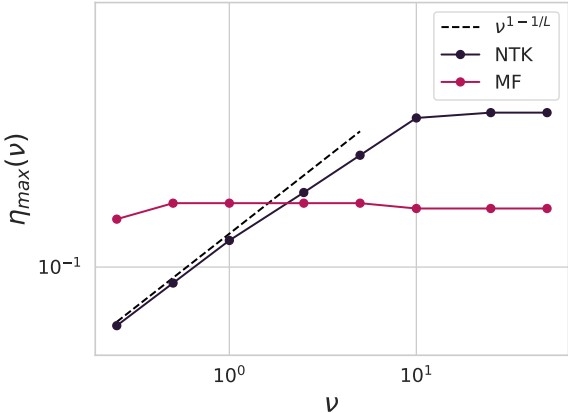

*Figure 8.* Maximal stable learning rate as a function of width $\nu$ for different parameterizations.

For very rich dynamics $\gamma_0 \gg 1$, the optimal learning rate approximately scales as

$$\eta_{\max} \sim \begin{cases} \mathcal{O}_\nu\left(1\right) & \text{MF (Fixed } \gamma_0) \\ \mathcal{O}_\nu\left(\nu^{1-\frac{1}{L}}\right) & \text{NTK (Fixed } \gamma_0\sqrt{\nu}), \end{cases} \tag{70}$$

based on a simple argument about the scale of the kernel. On the other hand, in the lazy learning regime $\gamma_0 \ll 1$, both MF and NTK parameterizations approach constant optimal learning rate as they approach linear models.

## C. Residual Networks

In this section, we consider training residual networks. To streamline the calculation in this section, we will first compute gradient descent on the population risk (incorporating random data can be handled by introducing the $\Delta(t)$ variables as in the previous sections). The key field definitions have the form

$$\boldsymbol{h}^0(t) = \boldsymbol{w}^\star - \frac{\sqrt{D}}{\gamma_0 N}\boldsymbol{W}^0(t)^\top \boldsymbol{g}^1(t)$$

$$\boldsymbol{h}^{\ell+1}(t) = \boldsymbol{h}^\ell(t) + \frac{\beta}{\sqrt{N}}\boldsymbol{W}^\ell(t)\boldsymbol{h}^\ell(t)$$

$$\boldsymbol{g}^L(t) = \boldsymbol{g}^L(0) + \eta\gamma_0 \sum_{t'<t} \boldsymbol{h}^L(t')$$

$$\boldsymbol{g}^\ell(t) = \boldsymbol{g}^{\ell+1}(t) + \frac{\beta}{\sqrt{N}}\boldsymbol{W}^\ell(t)^\top \boldsymbol{g}^{\ell+1}(t). \tag{71}$$

Introducing $\boldsymbol{\chi}^{\ell+1}(t) = \frac{1}{\sqrt{N}}\boldsymbol{W}^\ell(0)\boldsymbol{h}^\ell(t)$ and $\boldsymbol{\xi}^\ell(t) = \frac{1}{\sqrt{N}}\boldsymbol{W}^\ell(0)^\top \boldsymbol{g}^{\ell+1}(t)$, these obey the following DMFT single site dynamics

$$\chi^\ell(t) = u^\ell(t) + \sum_{t'<t} R_{hr}^{\ell-1}(t,t')g^\ell(t')$$

$$\xi^\ell(t) = r^\ell(t) + \sum_{t'<t} R_{gu}^{\ell+1}(t,t')h^\ell(t'). \tag{72}$$

The full gradient descent dynamics can thus be expanded as

We can solve simultaneously for correlation and response functions

$$h^0(t) = w^\star - r^0(t) - \sum_{t'<t}\left[\frac{1}{\gamma_0}R_{gu}^1(t,t') + \eta C_g^1(t,t')\right]h^0(t') \ , \ r^0(t) \sim \mathcal{GP}(0,\gamma_0^{-2}\nu^{-1}C_g^1) \tag{73}$$

$$h^1(t) = u^1(t) + \sum_{t'<t}\left[\frac{1}{\nu\gamma_0}R_{hr}^0(t,t') + \eta\gamma_0 C_h^0(t,t')\right]g^1(t') \ , \ u^1(t) \sim \mathcal{GP}(0,C_h^0)$$

$$h^{\ell+1}(t) = h^\ell(t) + \beta u^{\ell+1}(t) + \beta\sum_{t'<t}\left[R_{hr}^\ell(t,t') + \eta\gamma_0\beta C_h^\ell(t,t')\right]g^{\ell+1}(t') \ , \ u^{\ell+1}(t) \sim \mathcal{GP}(0,C_h^\ell)$$

$$g^L(t) = r^L(t) + \eta\gamma_0\sum_{t'<t}h^L(t') \ , \ r^L(t) \sim \mathcal{GP}(0,1)$$

$$g^\ell(t) = g^{\ell+1}(t) + \beta r^\ell(t) + \beta\sum_{t'<t}\left[R_{gu}^{\ell+1}(t,t') + \eta\gamma_0\beta C_g^{\ell+1}(t,t')\right]h^\ell(t') \ , \ r^\ell(t) \sim \mathcal{GP}(0,C_g^{\ell+1}) \tag{74}$$

To obtain the response functions $R_{hr}^\ell(t,t')$, $R_{gu}^\ell(t,t')$, we actually must compute all cross-layer response functions which form a closed system of equations as

$$\frac{\partial h^1(t)}{\partial u^k(t')} = \delta_{k,1}\delta(t-t') + \sum_{t''<t}\left[\frac{1}{\nu\gamma_0}R_{hr}^0(t,t'') + \eta\gamma_0 C_h^0(t,t'')\right]\frac{\partial g^1(t'')}{\partial u^1(t')} \ ,$$

$$\frac{\partial h^1(t)}{\partial r^k(t')} = \sum_{t''<t}\left[\frac{1}{\nu\gamma_0}R_{hr}^0(t,t'') + \eta\gamma_0 C_h^0(t,t'')\right]\frac{\partial g^1(t'')}{\partial r^k(t')} \ , \ k \in \{1,...,L\}$$

$$\frac{\partial h^{\ell+1}(t)}{\partial u^k(t')} = \frac{\partial h^\ell(t)}{\partial u^k(t')} + \beta\sum_{t''<t}\left[R_{hr}^\ell(t,t'') + \eta\gamma_0\beta C_h^\ell(t,t'')\right]\frac{\partial g^{\ell+1}(t'')}{\partial u^k(t')} \ , \ \ell \in \{1,...,L\}$$

$$\frac{\partial h^{\ell+1}(t)}{\partial r^k(t')} = \frac{\partial h^\ell(t)}{\partial r^k(t')} + \beta\sum_{t''<t}\left[R_{hr}^\ell(t,t'') + \eta\gamma_0\beta C_h^\ell(t,t'')\right]\frac{\partial g^{\ell+1}(t'')}{\partial r^k(t')} \ , \ \ell \in \{1,...,L\}$$

$$\frac{\partial g^L(t)}{\partial u^k(t')} = \eta\gamma_0\sum_{t''<t}\frac{\partial h^L(t'')}{\partial u^k(t')}$$

$$\frac{\partial g^L(t)}{\partial r^k(t')} = \delta_{k,L}\delta(t-t') + \eta\gamma_0\sum_{t''<t}\frac{\partial h^L(t'')}{\partial r^k(t')}$$

$$\frac{\partial g^\ell(t)}{\partial u^k(t')} = \frac{\partial g^{\ell+1}(t)}{\partial u^k(t')} + \beta\delta_{\ell,k}\delta(t-t') + \beta\sum_{t''<t}\left[R_{gu}^{\ell+1}(t,t'') + \eta\gamma_0\beta C_g^{\ell+1}(t,t'')\right]\frac{\partial h^\ell(t'')}{\partial u^k(t')}$$

$$\frac{\partial g^\ell(t)}{\partial r^k(t')} = \frac{\partial g^{\ell+1}(t)}{\partial r^k(t')} + \beta\delta_{\ell,k}\delta(t-t') + \beta\sum_{t''<t}\left[R_{gu}^{\ell+1}(t,t'') + \eta\gamma_0\beta C_g^{\ell+1}(t,t'')\right]\frac{\partial h^\ell(t'')}{\partial r^k(t')}$$

After these response functions have been solved for, we must solve for the correlation functions using the single site densities. Concretely, we use the above recursion to obtain the $\ell = k$ diagonal elements

$$R_{hr}^\ell(t,t') = \left\langle\frac{\partial h^\ell(t)}{\partial r^\ell(t')}\right\rangle \ , \ R_{gu}^\ell(t,t') = \left\langle\frac{\partial g^\ell(t)}{\partial u^\ell(t')}\right\rangle. \tag{75}$$

Using the above response functions and invoking the linearity of the dynaimcs, we have the following decompositions of the $h^\ell(t), g^\ell(t)$ fields

$$h^\ell(t) = \sum_{kt'} \left[ \frac{\partial h^\ell(t)}{\partial u^k(t')} u^k(t') + \frac{\partial h^\ell(t)}{\partial r^k(t')} r^k(t') \right] \tag{76}$$

$$g^\ell(t) = \sum_{kt'} \left[ \frac{\partial g^\ell(t)}{\partial u^k(t')} u^k(t') + \frac{\partial g^\ell(t)}{\partial r^k(t')} r^k(t') \right]. \tag{77}$$

We next compute the correlation and response functions $C_h^\ell(t, t')$ and $C_g^\ell(t, t')$ which have the form

$$C_h^\ell(t, t') = \sum_{kk'ss'} \frac{\partial h^\ell(t)}{\partial u^k(s)} \frac{\partial h^\ell(t')}{\partial u^k(s')} C_h^{k-1}(s, s') + \sum_{kk'ss'} \frac{\partial h^\ell(t)}{\partial r^k(s)} \frac{\partial h^\ell(t')}{\partial r^k(s')} C_g^{k+1}(s, s') \tag{78}$$

$$C_g^\ell(t, t') = \sum_{kk'ss'} \frac{\partial g^\ell(t)}{\partial u^k(s)} \frac{\partial g^\ell(t')}{\partial u^k(s')} C_h^{k-1}(s, s') + \sum_{kk'ss'} \frac{\partial g^\ell(t)}{\partial r^k(s)} \frac{\partial g^\ell(t')}{\partial r^k(s')} C_g^{k+1}(s, s') \tag{79}$$

where the base cases are determined by

$$C_h^0(t, t') = \sum_{ss'} \mathcal{H}(t, s) \mathcal{H}(t', s') \left[ 1 + \frac{1}{\nu \gamma_0^2} C_g^1(s, s') \right] \tag{80}$$

where the transfer function $\mathcal{H}(t, s)$ satisfies

$$\mathcal{H}(t, s) = \delta(t - s) + \sum_{t' < t} \left[ \frac{1}{\gamma_0} R_{gu}^1(t, t') + \eta C_g^1(t, t') \right] \mathcal{H}(t', s). \tag{81}$$

The test loss can finally be obtained from the $t, t$ diagonal entries

$$\mathcal{L}(t) = C_h^0(t, t). \tag{82}$$

We see that the formulas for the response functions depend on the correlation functions and vice versa. These must all be solved for simultaneously.

### C.1. The Large $L$ Limit

The infinite depth limit with $\beta = \beta_0/\sqrt{L}$ can be interpreted as a collection of ODEs for the response functions and a collection of ODEs for the correlation functions along the hidden layers.

First, we note that under the scaling $\beta = \beta_0/\sqrt{L}$, that for any $k > 1$, the responses scale as

$$\frac{\partial h^\ell(t)}{\partial u^k(t')} = \Theta\left(\frac{\beta_0}{\sqrt{L}}\right) , \quad \frac{\partial g^\ell(t)}{\partial u^k(t')} = \Theta\left(\frac{\beta_0}{\sqrt{L}}\right) \tag{83}$$

and similarly that for all $k < L$, the response functions for the $r^k$ fields scale as

$$\frac{\partial h^\ell(t)}{\partial r^k(t')} = \Theta\left(\frac{\beta_0}{\sqrt{L}}\right) , \quad \frac{\partial g^\ell(t)}{\partial r^k(t')} = \Theta\left(\frac{\beta_0}{\sqrt{L}}\right) \tag{84}$$

To take the large $L$ limit, we introduce a notion of "layer time"

$$\tau = \frac{\ell}{L} \in [0, 1] \tag{85}$$

and we recognize that the finite $L$ ResNet update equations approximate a SDE with a discretized process over $\tau \in [0, 1]$ with step size $d\tau \sim 1/L$. We define the limited

$$h(\tau, t) = \lim_{L \to \infty} h^\ell(t)|_{\ell = L\tau} , \quad g(\tau, t) = \lim_{L \to \infty} g^\ell(t)|_{\ell = L\tau} \tag{86}$$

The functions we aim to characterize in the limit are the limiting correlation functions

$$C_h^\ell(\tau, t, t') \equiv \lim_{L \to \infty} C_h^\ell(t, t')|_{\ell = L\tau} \ , \ C_g^\ell(\tau, t, t') \equiv \lim_{L \to \infty} C_g^\ell(t, t')|_{\ell = L\tau} \tag{87}$$

and the following $\Theta(1)$ limiting response functions

$$R_{hr}(\tau, t, t') \equiv \lim_{L \to \infty} \frac{\sqrt{L}}{\beta_0} R_{hr}^\ell(t, t')|_{\ell = L\tau} \ , \ R_{gu}(\tau, t, t') \equiv \lim_{L \to \infty} \frac{\sqrt{L}}{\beta_0} R_{gu}^\ell(t, t')|_{\ell = L\tau} \tag{88}$$

With these functions introduced, the single site equations in the "body" of the ResNet take the form of coupled SDEs

$$dh(\tau, t) = \beta_0 du(\tau, t) + \beta_0^2 d\tau \sum_{t' < t} [R_{hr}(\tau, t, t') + \eta\gamma_0 C_h(\tau, t, t')] g(\tau, t')$$

$$dg(\tau, t) = -\beta_0 dr(\tau, t) - \beta_0^2 d\tau \sum_{t' < t} [R_{gu}(\tau, t, t') + \eta\gamma_0 C_g(\tau, t, t')] h(\tau, t') \tag{89}$$

The noise sources in these SDEs have the correlation structure

$$\langle du(\tau, t) du(\tau', t') \rangle = \delta(\tau - \tau') C_h(\tau, t, t') d\tau d\tau' \ , \ \langle dr(\tau, t) dr(\tau', t') \rangle = \delta(\tau - \tau') C_g(\tau, t, t') d\tau d\tau' \tag{90}$$

The $h$ SDE is to be integrated forward in time from $\tau = 0$ to $\tau = 1$ and the $g$ SDE is to be integrated backwards in time from $\tau = 1$ to $\tau = 0$. The necessary response and correlation functions can be computed from this SDE as

$$R_{hr}(\tau, t, t') = \left\langle \frac{\partial h(\tau, t)}{\partial du(\tau, t')} \right\rangle \ , \ R_{gu}(\tau, t, t') = \left\langle \frac{\partial g(\tau, t)}{\partial dr(\tau, t')} \right\rangle$$

$$R_{hu}(\tau, t, t') = \left\langle \frac{\partial g(\tau, t)}{\partial du(\tau, t')} \right\rangle \ , \ R_{gr}(\tau, t, t') = \left\langle \frac{\partial g(\tau, t)}{\partial dr(\tau, t')} \right\rangle$$

$$C_h(\tau, t, t') = \langle h(\tau, t) h(\tau, t') \rangle \ , \ C_g(\tau, t, t') = \langle g(\tau, t) g(\tau, t') \rangle \tag{91}$$

However, we note that the single-layer response functions do not actually close, but rather, all possible pairs of layer-times must be considered, ie we must compute the full collection

$$R_{hr}(\tau, \tau', t, t') = \left\langle \frac{\partial h(\tau, t)}{\partial du(\tau', t')} \right\rangle \ , \ R_{gu}(\tau, \tau', t, t') = \left\langle \frac{\partial g(\tau, t)}{\partial dr(\tau', t')} \right\rangle \tag{92}$$

These four response functions form a set of ordinary differential equations

$$\partial_\tau R_{hu}(\tau, t, t') = \delta(t - t') + \beta_0 \sum_{t'' < t} [R_{hr}(\tau, t, t') + \eta\gamma_0 C_h(\tau, t, t')] R_{gu}(\tau, t'', t')$$

$$\partial_\tau R_{hr}(\tau, t, t') = \beta_0 \sum_{t'' < t} [R_{hr}(\tau, t, t') + \eta\gamma_0 C_h(\tau, t, t')] R_{gr}(\tau, t'', t')$$

$$\partial_\tau R_{gu}(\tau, t, t') = -\beta_0 \sum_{t'' < t} [R_{hr}(\tau, t, t') + \eta\gamma_0 C_h(\tau, t, t')] R_{hu}(\tau, t'', t')$$

$$\partial_\tau R_{gr}(\tau, t, t') = -\delta(t - t') - \beta_0 \sum_{t'' < t} [R_{hr}(\tau, t, t') + \eta\gamma_0 C_h(\tau, t, t')] R_{hr}(\tau, t'', t'). \tag{93}$$

Again from these response functions, we can again obtain the final correlation functions,

$$C_h(\tau, t, t') = \sum_{ss'} \int_0^1 d\tau' [R_{hu}(\tau, \tau', t, s) R_{hu}(\tau, \tau', t', s') C_h(\tau', s, s') + R_{hr}(\tau, \tau', t, s) R_{hr}(\tau, \tau', t', s') C_g(\tau', s, s')]$$

$$C_g(\tau, t, t') = \sum_{ss'} \int_0^1 d\tau' [R_{gu}(\tau, \tau', t, s) R_{gu}(\tau, \tau', t', s') C_h(\tau', s, s') + R_{gr}(\tau, \tau', t, s) R_{gr}(\tau, \tau', t', s') C_g(\tau', s, s')] \tag{94}$$

and the test loss dynamics have the form

$$\mathcal{L}(t) = C_h(0, t, t). \tag{95}$$

## D. Dynamics of Online SGD

In this section, we characterize discrete time online SGD where at each step $t$, a full batch of data $\boldsymbol{X}(t) \in \mathbb{R}^{B \times D}$ are sampled. This results in the following field definitions

$$\boldsymbol{h}^0(t) = \frac{\sqrt{D}}{B} \boldsymbol{X}(t)^\top \boldsymbol{\Delta}(t) \,, \ \boldsymbol{\Delta}(t) = \frac{1}{\sqrt{D}} \boldsymbol{X}(t) \boldsymbol{v}(t) + \sigma \boldsymbol{\epsilon}(t) \tag{96}$$

where all other variables are defined as before. Averaging over the samples drawn during SGD induces the following

$$h^0(t) = u^0(t) + v(t) \,, \ u^0(t) \sim \mathcal{N}\left(0, \frac{1}{\alpha_B} C_\Delta(t, t') \, \delta(t - t')\right) \tag{97}$$

$$\Delta(t) = u_\Delta(t) + \sigma \epsilon(t) \,, \ u_\Delta(t) \sim \mathcal{N}(0, C_v) \tag{98}$$

The key differences between online SGD and full batch GD training is

1. The noise $u^0$ is local in training steps (decorrelated across time)

2. There is no nonlocal response function $R_\Delta(t, t')$ that governs the buildup of the gap between test loss and train loss. Instead, the variance of $\langle h^0(t)^2 \rangle = C_v(t, t) + \frac{1}{\alpha_B} C_\Delta(t, t)$ is simply shifted by the SGD noise effect.

The rest of the DMFT equations for $\{h^\ell(t), g^\ell(t)\}_{\ell \in \{1, .., L\}}$ are the same as before as we provide below

### D.1. Full DMFT Equations for Online SGD

Below we provide the full set of DMFT equations for the online learning setting where batchsize $B$, width $N$ and dimension $D$ diverge proportionally

$$D, N, B \to \infty \,, \ B = \alpha D \,, \ N = \nu D. \tag{99}$$

In this limit, we obtain the following stochastic process for $\{v(t), \Delta(t), \{h^\ell(t)\}_{\ell=0}^L, \{g^\ell(t)\}_{L=1}^L\}$

$$v(t) = w^\star - r^0(t) - \sum_{t' < t} \left[\frac{1}{\gamma_0} R_{gu}^1(t, t') + \eta \gamma_0 C_g^1(t, t')\right] h^0(t') \,, \ r^0(t) \sim \mathcal{GP}\left(0, \frac{1}{\nu \gamma_0^2} G^1(t, t')\right)$$

$$\Delta(t) = u_\Delta(t) + \sigma \epsilon(t) \,, \ u_\Delta(t) \sim \mathcal{GP}\left(0, C_v(t.t')\right) \,, \ \epsilon(t) \sim \mathcal{N}(0, \delta(t - t'))$$

$$h^0(t) = u^0(t) + v(t) \,, \ u^0(t) \sim \mathcal{GP}\left(0, \frac{1}{\alpha_B} C_\Delta(t, t') \delta(t - t')\right)$$

$$h^1(t) = u^1(t) + \sum_{t' < t} \left[\frac{1}{\nu \gamma_0} R_{hr}^0(t, t') + \eta \gamma_0 C_h^0(t, t')\right] g^1(t') \,, \ u^1(t) \sim \mathcal{GP}(0, C_h^0(t, t'))$$

$$h^\ell(t) = u^\ell(t) + \sum_{t' < t} \left[R_{hr}^{\ell-1}(t, t') + \eta \gamma_0 C_h^{\ell-1}(t, t')\right] g^\ell(t') \,, \ u^\ell(t) \sim \mathcal{GP}(0, C_h^{\ell-1}(t, t'))$$

$$g^\ell(t) = r^\ell(t) + \sum_{t' < t} \left[R_{gu}^{\ell+1}(t, t') + \eta \gamma_0 C_g^{\ell+1}(t, t')\right] h^\ell(t') \,, \ r^\ell(t) \sim \mathcal{GP}(0, C_g^{\ell+1}(t, t')) \tag{100}$$

where as before the response functions are defined as

$$R_{vu}^0(t, t') = \left\langle \frac{\partial v(t)}{\partial u^0(t')} \right\rangle \,, \ R_{hr}^0(t, t') = \left\langle \frac{\partial h^0(t)}{\partial r^0(t')} \right\rangle$$

$$R_{hr}^\ell(t, t') = \left\langle \frac{\partial h^\ell(t)}{\partial r^\ell(t')} \right\rangle \,, \ R_{gu}^\ell(t, t') = \left\langle \frac{\partial g^\ell(t)}{\partial u^\ell(t')} \right\rangle \,, \tag{101}$$

and the correlation functions are defined as usual

$$C_v(t, t') = \langle v(t) v(t') \rangle \,, \ C_\Delta(t, t') = \langle \Delta(t) \Delta(t') \rangle$$

$$C_h^\ell(t, t') = \langle h^\ell(t) h^\ell(t') \rangle \,, \ C_g^\ell(t, t') = \langle g^\ell(t) g^\ell(t') \rangle. \tag{102}$$

## D.2. Analysis of Early Portion of Online SGD

From our theory and simulations, we see that for sufficiently large learning rates and sufficiently small width and batch size, the test loss can initially *increase* with iterations $t$ before subsequently recovering and converging at later time. In this section, we analyze this effect as an approximation of the early loss dynamics. Neglecting the memory terms that buildup over longer training times, we make the following approximations for small $t$

$$
\begin{aligned}
R^1_{gu}(t,t') &\approx \eta\gamma_0 L\Theta(t-t') \\
G^1(t+1,t+1) - G^1(t,t) &\approx [G^2(t+1,t+1) - G^2(t,t)] + 4\eta^2 L^2\gamma_0^2 C^0_h(t,t) \\
&\approx \eta^2 C^0_h(t,t)\sum_{k=1}^{L}k^2 = \frac{\eta^2\gamma_0^2 L(L+1)(2L+1)}{6}C^0_h(t,t)
\end{aligned}
\tag{103}
$$

which can be justified by working out the early dynamics for the response functions at small time. Under this approximation, we derive the following recursion for the test loss $C_v(t) = \langle v(t)^2 \rangle$, which has the form

$$
C_v(t+1) \approx \left[ (1-L\eta)^2 + \frac{\eta^2 L^2}{\alpha_B} + \frac{\eta^2 L(L+1)(2L+1)}{6\nu} + \frac{\eta^2 L(L+1)(2L+1)}{6\nu\alpha_B} \right] C_v(t).
\tag{104}
$$

We verify that this does accurately predict the first few steps of the loss dynamics (and captures the early blowup) of deep linear networks in Figure 9. Under this early time approximation, the loss will thus increase if

$$
\eta L > \frac{2}{1 + \frac{1}{\alpha_B} + \frac{(L+1)(2L+1)}{6L\nu} + \frac{(L+1)(2L+1)}{6L\alpha_B\nu}}.
\tag{105}
$$

Indeed from simulations, we see that the early dynamics diverge for sufficiently small $\alpha_B$ and $\nu$.

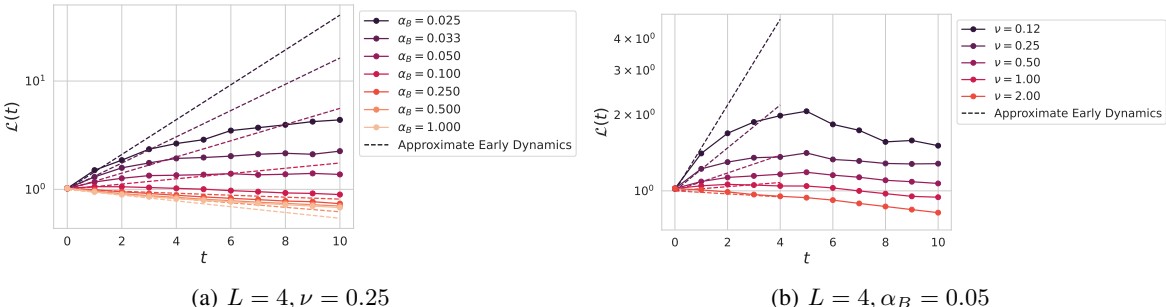

$$\text{(a) } L = 4, \nu = 0.25 \qquad\qquad\qquad \text{(b) } L = 4, \alpha_B = 0.05$$

*Figure 9.* Early dynamics of the online SGD can exhibit an initial blowup followed by recovery if $\gamma_0 > 0$. The initial blowup is approximately $\mathcal{L}(t) \approx \left[ (1-\eta L)^2 + \eta^2 L^2 \left( \alpha_B^{-1} + \nu^{-1} + 2\alpha_B^{-1}\nu^{-1} \right) \right]^t$, which is a function of learning rate $\eta$, depth $L$, batchsize $\alpha_B$ and width $\nu$. Small width or small batchsize can make this effect more pronounced.

# E. Online SGD with Structured (Dimension Free) Data

In this section we discuss the effect of structured data on the dynamics. In this setting, we allow $x \sim \mathcal{N}(0, \Lambda)$ with $\text{Tr}\Lambda < \infty$ in the $D \to \infty$ limit. Without loss of generality, we let $\Lambda$ be diagonal with entries $\Lambda_{kk} = \lambda_k$ but allow for arbitrary $w^\star$. In this case, the proper definition of the variables $h^0$ and $\Delta(t)$ are

$$
\begin{aligned}
h^0(t) &= \frac{1}{B}X(t)^\top\Delta(t)\,, \ \Delta(t) = X(t)v(t) + \sigma\epsilon(t) \\
h^1(t) &= W^0(t)h^0(t)\,, \ v(t) = w^\star - \frac{1}{N\gamma_0}W^0(t)^\top g^1(t) + \frac{1}{N\gamma_0}W^0(0)^\top g^1(0)
\end{aligned}
\tag{106}
$$

With centering, this gives the following mean field description

$$h_k^0(t) = u_k^0(t) + \lambda_k v_k(t) \, , \; \Delta(t) = u_\Delta(t) + \sigma\epsilon(t) \, , \; u_k^0(t) \sim \mathcal{N}\left(0, \frac{\lambda_k}{B}C_\Delta\delta(t-t')\right) \, , \; u_\Delta(t) \sim \mathcal{N}(0, C_v)$$

$$v_k(t) = w_k^\star - r_k^0(t) + r_k^0(0) - \sum_{t'<t}\left[\frac{1}{\gamma_0}R_{gu}^1(t,t') + \eta C_g^1(t,t')\right]h_k^0(t') \, , \; r_k^0(t) \sim \mathcal{N}\left(0, \frac{1}{N\gamma_0^2}C_g^1\right) \quad (107)$$

The definition of the correlation functions for these fields changes to

$$C_v(t,t') = \sum_{k=1}^\infty \lambda_k \langle v_k(t)v_k(t')\rangle \, , \; C_h^0(t,t') = \sum_{k=1}^\infty \langle h_k^0(t)h_k^0(t')\rangle \quad (108)$$

while the response function for the $h^0$ and $v$ variables change to

$$R_{hr}^0(t,t') = \sum_{k=1}^\infty \left\langle \frac{\partial h_k^0(t)}{\partial r_k^0(t')}\right\rangle \, , \; R_{vu}(t,t') = \sum_{k=1}^\infty \lambda_k \left\langle \frac{\partial v_k(t)}{\partial u_k^0(t')}\right\rangle \quad (109)$$

which are both finite due to the trace class condition $\sum_k \lambda_k < \infty$. To express the test loss, we can introduce the $k$-th mode's transfer function $\mathcal{H}_k(t,t')$ which has the following definition

$$v_k(t) = \sum_{t'<t}\mathcal{H}_k(t,t')\left[w_k^\star - r_k^0(t') - \sum_{t''<t'}\left[\gamma_0^{-1}R_{gu}^1(t',t'') + C_g^1(t',t'')\right]u_k^0(t'')\right] \quad (110)$$

$$\mathcal{H}_k(t,t') \equiv \delta(t-t') - \lambda_k\sum_{t''<t}\left[\gamma_0^{-1}R_{gu}^1(t,t'') + \eta C_g^1(t,t'')\right]\mathcal{H}_k(t'',t'). \quad (111)$$

The other hidden layers of the network have the same single site dynamics as before,

$$h^1(t) = u^1(t) + \sum_{t'<t}\left[\frac{1}{N\gamma_0}R_{hr}^0(t,t') + \eta\gamma_0 C_h^0(t,t')\right]g^1(t') \, , \; u^1(t) \sim \mathcal{N}(0, C_h^0)$$

$$h^\ell(t) = u^\ell(t) + \sum_{t'<t}\left[R_{hr}^{\ell-1}(t,t') + \eta\gamma_0 C_h^{\ell-1}(t,t')\right]g^\ell(t') \, , \; u^\ell(t) \sim \mathcal{N}(0, C_h^{\ell-1})$$

$$g^L(t) = r^L(t) + \eta\gamma_0\sum_{t'<t}h^L(t')$$

$$g^\ell(t) = r^\ell(t) + \sum_{t'<t}\left[R_{gu}^{\ell+1}(t,t') + \eta\gamma_0 C_g^{\ell+1}(t,t')\right]h^\ell(t') \, , \; u^\ell(t) \sim \mathcal{N}(0, C_g^{\ell+1})$$

### E.1. Scaling Laws for Easy and Hard Tasks

We now examine the scaling law behavior with respect to time $t$ for the $N, B \to \infty$ limit. We also study the small learning rate regime, where we can study a continuous time gradient flow dynamics

$$\partial_t v_k(t) \approx -\lambda_k K(t)v_k(t) \quad (112)$$

where $K(t) \approx C_g^1(t,t) + \gamma_0^{-1}R_{gu}^1(t,t)$ is related to the *Neural Tangent Kernel* for the model (Jacot et al., 2018). We make the following ansatz for the small $\gamma_0$ dynamics for $K(t)$

$$K(t) \sim 1 + \gamma_0^2 t^{\chi-1} \, , \; \gamma_0 \to 0 \quad (113)$$

where the above is understood to disregard prefactors constants. Following general arguments of Bordelon et al. (2024b), we must try identifying solutions to the self consistent equation

$$\partial_t v_k(t) = -\lambda_k[1 + \gamma_0^2 t^{\chi-1}]v_k(t) \implies v_k(t) = \exp\left(-\lambda_k[t + \gamma_0^2 t^\chi]\right)w_k^\star \quad (114)$$

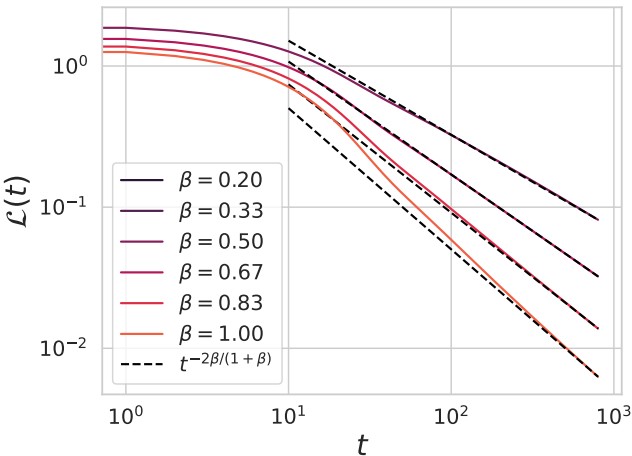

*Figure 10.* Scaling laws for the hard task regime are well predicted by the adjusted exponent.

Following the self-consistency argument, we find that the kernel must scale as $K(t) \sim t^{1-\beta\chi}$. Combining this with the assumption implies that

$$\chi = \frac{2}{1+\beta} \; , \; \forall \beta < 1. \tag{115}$$

This implies the following loss scalings

$$\mathcal{L}(t) \sim \begin{cases} t^{-\beta} & \beta > 1 \\ t^{-\frac{2\beta}{1+\beta}} & \beta < 1 \end{cases} \tag{116}$$

which we show in Figure 10. The case where $\beta < 1$ is known as the "hard task" regime while $\beta > 1$ is the "easy task" regime for linear networks trained on powerlaw features.

## F. Different Relative Widths

We can also handle the case where hidden widths $\{N_\ell\}_{\ell=1}^L$ differ across layers

$$f(\boldsymbol{x}) = \frac{\sqrt{D}}{N_L \gamma_0} \boldsymbol{w}^L \prod_{\ell=1}^{L-1} \left( \frac{1}{\sqrt{N_\ell}} \boldsymbol{W}^\ell \right) \left( \frac{1}{\sqrt{D}} \boldsymbol{W}^0 \right) \boldsymbol{x}. \tag{117}$$

In this case, the proportional limit of interest is

$$\alpha = \frac{P}{D} \; , \; \nu_\ell = \frac{N_\ell}{D}. \tag{118}$$

We can straightforwardly work out the limit using the same techniques as before.

## G. Multiple Output Channels

The case of multiple (but $\mathcal{O}(1)$) output channels $N_c$ can also be handled within our formalism. The cost of this analysis is to track correlation and response functions across all possible pairs of output channels as well as correlations across timesteps. We let the output channel $c$ have readout $\boldsymbol{w}_c^L$ as before

$$f_c(\boldsymbol{x}) = \frac{1}{N\gamma_0} \boldsymbol{w}_c^L \prod_{\ell=1}^{L-1} \left( \frac{1}{\sqrt{N}} \boldsymbol{W}^\ell \right) \left( \frac{1}{\sqrt{D}} \boldsymbol{W}^0 \right) \boldsymbol{x} \tag{119}$$

In this case, we assume a collection of target vectors $\boldsymbol{w}_c^\star$ for $c \in \{1, ..., N_c\}$. In this case, the test error is governed by the following weight discrepancy

$$\boldsymbol{v}_c(t) = \boldsymbol{w}_c^\star - \frac{1}{N\gamma_0}\boldsymbol{W}^0(t)^\top \boldsymbol{g}_c^1(t)$$

$$\boldsymbol{g}_c^\ell(t) = \frac{1}{\sqrt{N}}\boldsymbol{W}^\ell(t)^\top \boldsymbol{g}_c^{\ell+1}(t) \,, \; \boldsymbol{g}_c^L(t) = \boldsymbol{w}_c^L(t) \tag{120}$$

The dynamics for the weights are

$$\boldsymbol{W}^0(t+1) = \boldsymbol{W}^0(t) + \frac{\eta\gamma_0}{\sqrt{D}}\sum_{c=1}^{N_c}\boldsymbol{g}_c^1(t)\boldsymbol{h}_c^0(t)^\top$$

$$\boldsymbol{W}^\ell(t+1) = \boldsymbol{W}^\ell(t) - \frac{\eta\gamma_0}{\sqrt{N}}\sum_{c=1}^{N_c}\boldsymbol{g}_c^{\ell+1}(t)\boldsymbol{h}_c^\ell(t)^\top$$

$$\boldsymbol{w}_c^L(t+1) = \boldsymbol{w}_c^L(t) + \eta\gamma_0\boldsymbol{h}_c(t) \tag{121}$$

where the variables $\boldsymbol{h}_c^\ell(t)$ are defined as

$$\boldsymbol{h}_c^{\ell+1}(t) = \frac{1}{\sqrt{N}}\boldsymbol{W}^\ell(t)\boldsymbol{h}_c^\ell(t) \,, \; \boldsymbol{h}_c^0(t) = \frac{\sqrt{D}}{P}\sum_{\mu=1}^P \Delta_{\mu,c}(t)\boldsymbol{x}_\mu \tag{122}$$

Expanding out the dynamics we find

$$\boldsymbol{h}_c^\ell(t) = \boldsymbol{\chi}_c^\ell(t) + \eta\gamma_0\sum_{t'<t}\sum_{c'}C_{h,cc'}^{\ell-1}(t,t')\boldsymbol{g}_{c'}^\ell(t')$$

$$\boldsymbol{g}_c^\ell(t) = \boldsymbol{\xi}_c^\ell(t) + \eta\gamma_0\sum_{t'<t}\sum_{c'}C_{g,cc'}^{\ell+1}(t,t')\boldsymbol{h}_{c'}^\ell(t') \tag{123}$$

As before we can carry out the DMFT computation but the order parameters that we need to track include cross-output channel correlations

$$C_{v,cc'}^\ell(t,t') = \langle v_c(t)v_{c'}(t')\rangle \,, \; C_{\Delta,cc'}^\ell(t,t') = \langle \Delta_c(t)\Delta_{c'}(t')\rangle$$
$$C_{h,cc'}^\ell(t,t') = \langle \boldsymbol{h}_c^\ell(t)\boldsymbol{h}_{c'}^\ell(t')\rangle \,, \; C_{g,cc'}^\ell(t,t') = \langle \boldsymbol{g}_c^\ell(t)\boldsymbol{g}_{c'}^\ell(t')\rangle \tag{124}$$

as well as cross channel response functions

$$R_{vu,cc'}(t,t') = \left\langle \frac{\delta v_c(t)}{\delta u_{c'}^0(t')}\right\rangle \,, \; R_{\Delta,cc'}(t,t') = \left\langle \frac{\delta\Delta_c(t)}{\delta u_{\Delta,c'}(t')}\right\rangle$$

$$R_{hr,cc'}^\ell(t,t') = \left\langle \frac{\delta \boldsymbol{h}_c^\ell(t)}{\delta r_{c'}^\ell(t')}\right\rangle \,, \; R_{gu,cc'}^\ell(t,t') = \left\langle \frac{\delta \boldsymbol{g}_c^\ell(t)}{\delta u_{c'}^\ell(t')}\right\rangle \tag{125}$$

These correlations and response functions close in the obvious generalization of the scalar output case.

## H. Comparing Computational Costs

In this section we compare the cost of training a width $N$ and depth $L$ network on $P$ data with full batch GD to integrating the DMFT equations in different settings.

|  | Finite Network | Linear DMFT | Res. DMFT | Non-Data Averaged Res. DMFT |
|---|:---:|:---:|:---:|:---:|
| Memory | $N^2 L$ | $LT^2$ | $L^2 T^2$ | $L^2 T^2 P^2$ |
| Compute | $N^2 LPT$ | $LT^3$ | $L^2 T^3$ | $L^2 T^3 P^3$ |

*Table 1.* Memory and computational cost to describe full batch gradient descent training of width $N$ depth $L$ network trained on dataset of size $P$ for $T$ steps. The non-data averaged residual DMFT refers to the cost of solving the equations in (Bordelon et al., 2024c).

