# OpenReview forum: "Deep Linear Network Training Dynamics from Random Initialization: Data, Width, Depth, and Hyperparameter Transfer"
_ICML.cc/2025/Conference — ICML 2025 poster_

### Official Review · Reviewer_ZpVM · 2025-03-06

**Overall Recommendation:** 2

**Summary:**

This work provides a theoretical analysis of gradient descent dynamics in deep linear networks trained at large widths from random initialisation. Specifically, gradient descent dynamics, hyper-parameter transfer effects and asymptotic descriptions for deep networks were analysed and discussed.

**Claims And Evidence:**

Claims are backed by rigorous theoretical derivations.

**Essential References Not Discussed:**

Not applicable.

**Experimental Designs Or Analyses:**

Experimental experiments are relatively limited.

**Methods And Evaluation Criteria:**

The evaluation is purely theoretical and does not include standard deep learning benchmarks.

**Other Comments Or Suggestions:**

None.

**Other Strengths And Weaknesses:**

The paper provides strong theoretical insights, but lacks extensive empirical validation on real-world datasets.

**Questions For Authors:**

No specific questions.

**Relation To Broader Scientific Literature:**

Insights on width-depth interactions and hyper-parameter transfer are quite novel, but focusing linear networks limits its scope and application in real cases.

**Theoretical Claims:**

Mathematical formulations are elegant but not carefully verified.

---

> ### Author Rebuttal · Authors · 2025-03-30
>
> We thank the reviewer for appreciating our theoretical contributions and the novelty of attempting to capture the hyperparameter transfer effect.
>
> ### Methods and Evaluation Criteria
>
> *The evaluation is purely theoretical and does not include standard deep learning benchmarks.*
>
> The purpose of this paper is primarily theoretical and focuses on a linear networks, rather than an experimental paper purporting to perform well on benchmarks.
>
>
> ### Theoretical Claims
>
> *Mathematical formulations are elegant but not carefully verified.*
>
> We provide a derivation of our results in the Appendix using techniques from statistical mechanics. We also verify our equations under the conditions of our theory (randomly initialized deep linear networks trained on random data) using simulations.
>
>
> ### Relation To Broader Scientific Literature:
>
> *Insights on width-depth interactions and hyper-parameter transfer are quite novel, but focusing linear networks limits its scope and application in real cases.*
>
> Our focus on linear networks was motivated by finding the **simplest possible model** that exhibits the phenomenon of interest (which in this case is the learning rate transfer effect).
>
> ### Other Strengths And Weaknesses
>
> *The paper provides strong theoretical insights, but lacks extensive empirical validation on real-world datasets.*
>
> Since we are focused on the dynamics of **linear networks**, the generalization on most real-world datasets would not be very good. However, we can attempt to include an experiment on a simple task like MNIST, which due to its power law covariance spectrum (see Figure 6a here https://arxiv.org/abs/2409.17858) would likely look similar to our Figure 6c.

---

### Official Review · Reviewer_HCdG · 2025-03-08

**Overall Recommendation:** 3

**Summary:**

This work theoritically characterizes the gradient descent dynamics in deep linear networks in the asymptotic limit of infinite data and width of the network. They study the limiting behaviour of both deep linear network and residual deep network for both isotropic data and data with power-law covariance and studies the effect of depth, richness, batch-size and hyper-parameter transfer.

**Claims And Evidence:**

Yes, the claims and evidence are convincing. However, some parameters remains undefined, so it is difficult for the reader to grasp the essence of the results.

**Essential References Not Discussed:**

Although there are tons of works on deep linear networks, I think the current refernces are sufficient for the paper.

**Experimental Designs Or Analyses:**

Yes, they are valid.

**Methods And Evaluation Criteria:**

It is mostly linear networks and labels genrated by linear models.

**Other Comments Or Suggestions:**

1) Please define $\mu P$.
2) In section-2.1, dynamics are studies in the aymptotic limit of P,D,N. However, why is there a further limit on $\alpha$ and $\alpha_{B}$. This imit was unclear.

## update after rebuttal

I have read the rebuttal and decided to stick with my score of weak accept.

**Other Strengths And Weaknesses:**

4) The effect of learning rate seemed to be missing from the result. What about the alignment between singular vector for each layer? Does the traditional result of implicit bias of gradient flow and descent captureed here? It seemed no from the current result. Discussing this in details would be more useful.

**Questions For Authors:**

Please answer points 1,2,3,4.

**Relation To Broader Scientific Literature:**

Studying the dynamics of deep linear networks is an important first step to understand deep neural networks. This work should be highly relevant.

**Theoretical Claims:**

The proofs and the theory looks mostly correct to me except some minor questions:

1) $\gamma_{0}$ is defined nowhere in the text and it is hard to grasp the meaning of this quanitity alhtough i see several results depend on it.
2) In equation-10, no effect of learning rate or intializations scale is captured. It is well known that rich regime is mostly captured with small initialization and small learning rate, but this result was not seen from the theory result.
3) Linear networks are known to exhibit saddle to saddle behaviour (for rich dynamics)...However, is it true that this dynamics can't be captured by isotropic random data?

---

> ### Author Rebuttal · Authors · 2025-03-30
>
> We thank the reviewer for their careful reading and detailed comments and questions. We address the main concerns below.
>
> ### Theoretical Claims
>
> *$\gamma_0$ is defined nowhere in the text and it is hard to grasp the meaning of this quantity alhtough i see several results depend on it.*
>
> We will define this hyperparameter more clearly. Mechanically it is present in the definition of the predictor $f = \frac{1}{\gamma_0} w^L ... W^0 x$. This scalar basically controls the laziness/richness of the learning dynamics with $\gamma_0 \to 0$ giving the lazy (kernel) regime. For $\gamma_0 \gg 0$, the dynamics are far from the kernel regime. We have added more explanation of this and also shown that $\gamma_0 \to 0$ limit gives the kernel regime. The dependence of $\gamma_0$ on network width also defines the difference between NTK parameterization and mean-field/$\mu$P scaling.
>
> *In equation-10, no effect of learning rate or intializations scale is captured. It is well known that rich regime is mostly captured with small initialization and small learning rate, but this result was not seen from the theory result.*
>
> Equation 10 depends implicitly on the hyperparameter $\gamma_0$ which we are using to control the laziness/richness. We can alternatively manipulate the initail variance of the weight entries which would have a similar effect.
>
> *Linear networks are known to exhibit saddle to saddle behaviour (for rich dynamics)...However, is it true that this dynamics can't be captured by isotropic random data?*
>
> Our dynamics would start at a *single saddle point* for $\gamma_0 \gg 1$ limit with the learning rate scaled down. We could induce **multiple saddles** (saddle-to-saddle behaviors) in a multiple output channel setting with isotropic features, especially if we also allow for small initial weights. We have equations for this setting (Appendix G) but have not numerically integrated them yet. Indeed, in the small initialization regime this is exactly the setting of the work of Saxe et al 2014 https://arxiv.org/abs/1312.6120.
>
> *The effect of learning rate seemed to be missing from the result. What about the alignment between singular vector for each layer? Does the traditional result of implicit bias of gradient flow and descent captureed here? It seemed no from the current result. Discussing this in details would be more useful.*
>
> Our theoretical equations depend directly on the learning rate such as in equation 11 and equation 67.
>
> Our theory, since it applies to non-negligible random initialization, does not demand perfect alignment between adjacent weight matrices. However, in the rich regime, the alignment does tend to increase.  Equation 25 reveals that
>
> $$W^\ell(t) = W^\ell(0) + \frac{\eta\gamma_0}{\sqrt N}\sum_{t'<t} g^{\ell+1}(t') h^\ell(t')^\top$$
>
> The first term is random and static, while the second term improves alignment of $W^\ell$'s left singular vectors with the $\{ g^\ell(t) \}$ vectors which are
> \begin{align}
> g^{\ell}(t) = \left[ \frac{1}{\sqrt{N}} W^{\ell}(t) \right]^\top ... w^L(t)
> \end{align}
> If the random initialization were negligible compared to the second term, then all weights would align and become low rank, consistent with prior works. However, our theory can flexibly handle large random initialization in either lazy or rich regimes (for arbitrary values of $\gamma_0$).
>
> ### Other Comments or Suggestions
>
> *Please define $\mu$P*
>
> We have added a definition of $\mu$P as "maximum update parameterization", a term introduced in https://arxiv.org/abs/2011.14522. We will add this definition.
>
> *In section-2.1, dynamics are studies in the aymptotic limit of P,D,N. However, why is there a further limit on $\alpha$ and $\alpha_B$. This limit was unclear.*
>
> We apologize that our exposition in this section was unclear to the reader. We will try explaining it more clearly in section 2.1. We basically analyze two settings
>
> 1. Full batch gradient descent with a fixed random dataset of size $P$ in a proportional scaling regime $P,D,N\to\infty$ with fixed ratios $P/D = \alpha$ and $N/D = \nu$.
> 2. Online stochastic gradient descent where at each step a batch of size $B$ is sampled and used to estimate the gradient. We look at $B,N,D \to \infty$ with fixed ratios $B/D = \alpha_B$ and $N/D = \nu$.
>
> We contrast these two settings in Figure 3, where Figure 3 (a,b) is in the first setting and Figure 3  (c,d) is in the second setting.

---

### Official Review · Reviewer_4bAU · 2025-03-10

**Overall Recommendation:** 3

**Summary:**

This paper develops a DMFT based theory for deep linear networks (with and without residual connections) in GD and SGD settings. The authors show that the theory captures the effect of initialization, dataset correlations, width and depth. Moreover, they show hyperparameter transfer with width and depth.

**Claims And Evidence:**

Yes. The claims made in the paper are supported by evidence. Specifically, the figures validate the theory.

**Essential References Not Discussed:**

The essential references are discussed.

**Experimental Designs Or Analyses:**

Yes. The experimental designs are sound.

**Methods And Evaluation Criteria:**

Yes. The methods and evaluation make sense.

**Other Comments Or Suggestions:**

Comments:

* There is a typo in Equation 9, it should be $W^0$ and not $W^1$.
* Reference for Appendix is missing on line 150.

Suggestions:

* It would be helpful to provide further details for Sections 4 and 5 in the Appendix.

**Other Strengths And Weaknesses:**

Strengths

* The paper is clearly written and is easy to follow

* The insights on the complexity of data, depth and width are insightful

Weaknesses

* To the best of my knowledge, the hyperparameter transfer results are known in prior literature in much more complex settings.

* While the Section 3 results are clearly discussed in the Appendix, I found the details for Section 4 and 5 to be sparse in the Appendix. In Appendix C, the authors mention: "After these response functions have been solved for, we must solve for the correlation functions using the single site densities. This gives a closed form prediction for the loss dynamics." but do not provide details. Similarly, it would be helpful to expand on Appendix E details on structured data.

**Questions For Authors:**

Questions:

* Can the authors clarify if they have any additional insights into hyperparameter transfer compared to prior works?

* Do the authors understand why does the loss increase initially in Figure 3 (c, d)?

**Relation To Broader Scientific Literature:**

This work adds to the ongoing research on understanding the neural networks by focusing on deep linear networks (with residual connections). The work provides insights into the effect of data complexity, width, depth, and random initialization.

**Theoretical Claims:**

Yes. I have verified all the theoretical details in the main text and Appendix A, while skimming the remaining Appendices for soundness.

---

> ### Author Rebuttal · Authors · 2025-03-30
>
> We thank the reviewer for their careful reading and their support. Below we address the weaknesses.
>
> ### Weaknesses
>
> **hyperparameter transfer results are known in prior literature in much more complex settings**
>
> While there are already several cases where hyperparameter transfer are documented (including in complicated architectures like transformers), we were seeking the simplest possible setting which could be theoretically analyzed. To capture hyperparameter transfer effects one needs a setting where (1) model can exit the kernel regime and (2) wider models perform better. The simplest model we could identify with these properties was **randomly initialized deep linear networks**.
>
> **Appendix Explanations Sparse**
>
> We thank the reviewer for this comment. We have made the Appendix sections more detailed and provide the closed form set of equations for the correlation and response functions from the single-site equations. The train and test losses can be computed from the correlation functions $C_v$ and $C_\Delta$.
>
> ### Other comments
>
> Thank you for finding these typos and missing links. We have fixed these.
>
> *It would be helpful to provide further details for Sections 4 and 5 in the Appendix.*
>
> We have included additional details to derive the main results of section 4 and 5.
>
>
> ### Questions
>
> *Can the authors clarify if they have any additional insights into hyperparameter transfer compared to prior works?*
>
> Our main result for hyperparameter transfer is that in $\mu$P scaling, the finite width effects accumulate as a combination of effective noise and bias in the dynamics from finite $\nu  = N/D$ while the feature updates are approximately independent of $\nu$ (see equation 14).
>
> *Do the authors understand why does the loss increase initially in Figure 3 (c, d)?*
>
> Good question! This initial loss increase is driven by the variance in the predictor from SGD noise (small $\alpha_B$) and small width (small $\nu$). This can be seen from an analysis of the early portion of the DMFT equations where, for the first few steps of training the test loss can be approximated as
>
> $$\mathcal L(t+1) \approx \left[(1-\eta)^2 + \frac{\eta^2}{\alpha_B} + \frac{\eta^2}{\nu} \right] \mathcal{L}(t)$$
>
> The loss will exponentially increase at early times provided that
> \begin{align}
>     \eta > \frac{2}{1+\frac{1}{\alpha_B} + \frac{1}{\nu}} .
> \end{align}
>
> Indeed from the simulations we see that for sufficiently large $\alpha_B$ and $\nu$ this initial increase disappears.

---

> > ### Comment · Reviewer_4bAU · 2025-04-03
> >
> > I thank the authors for the clarifications. I will keep my score.

---

### Official Review · Reviewer_pEUF · 2025-03-13

**Overall Recommendation:** 4

**Summary:**

The authors analyze several models of deep linear networks (FCNs, ResNets) trained on Gaussian (iid, and power-law covariance) data with noisy Gradient Descent, focusing on hyperparameter transfer between small and large models. They develop a DMFT formulation of the problem which can accommodate finite width, finite learning effects, and dataset average. This results in a series of non-linear saddle point equations on kernels and gradient kernels which are solved numerically. The numerical solutions agree well with simulations of actual neural networks. At least in muP scaling, they argue that the optimal learning rate transfers properly across widths while the loss changes.

**Claims And Evidence:**

I found several issues with the manuscript.

The papers claim to give a theoretical result. However, since theoretical results are delivered implicitly via high-dimensional non-linear equations that require a numerical solver, it is not clear what new understanding one has gained.

A second issue is that optimal learning rate does in fact shift in their setting even in muP. The shift in optimal value itself seems somewhat larger than in Ref. https://arxiv.org/pdf/2203.03466 however the sharpness of the loss makes it so that, in figure 2b, choosing the learning rate based on v=0.5 and transferring to v=5, would incur an order of magnitude change to the loss compared to its optimal value.

A third issue is the concentration of the kernel in NTK scaling. While the authors arrive at an action which has D in front of an seemingly $D$ independent action for the kernels, the orders parameters involved have data indices and hence scale with D. In such circumstances, the saddle point is not clearly justified. A simple example is the Wishart Distribution, wherein by normalizing such that the matrix elements average to the identity, one has a similar structure of probability. However, taking a saddle point would yield a delta function distribution of eigenvalues, whereas in fact the width of the eigenvalue distribution is similar to its average. In muP scaling this issue should go away.

**Essential References Not Discussed:**

Nothing essential.

**Experimental Designs Or Analyses:**

The experiments look sufficient, apart from the previous comment about the sharpness of the loss.

**Methods And Evaluation Criteria:**

See above.

**Other Comments Or Suggestions:**

No.

**Other Strengths And Weaknesses:**

Strengths:

1. The authors develop a DMFT formalism for deep learning networks which accounts for data-averaging and finite learning rate effects.
2. Although not entirely clear from the manuscript, which doesn't give away absolute scales of D, they seem to push the numerical envelope of solving DMFT equation further.
3. The author provides a toy setting which, despite being too complicated to being solved analytically at the moment, may serves as a stepping stone to future analytical research in this important subfield.

Weaknesses:

1. Presentation: The papers claim to give a theoretical result. However, since the theoretical results are delivered implicitly via high-dimensional non-linear equations that require a numerical solver, it is not clear what new understanding one has gained.

2. Deficiencies of the toy setting: A second issue is that the optimal learning rate does in fact shift in their setting even in muP. The shift in optimal value itself seems somewhat larger than in Ref. https://arxiv.org/pdf/2203.03466 however the sharpness of the loss makes it so that, in figure 2b, choosing the learning rate based on v=0.5 and transferring to v=5, would incur an order of magnitude change to the loss compared to its optimal value.

3. Correctness of derivation for NTK scaling. While the authors arrive at an action which has D in front of an seemingly $D$ independent action for the kernels, the orders parameters involved have data indices and hence scale with D. In such circumstances, the saddle point is not clearly justified. A simple example is the Wishart Distribution, wherein by normalizing such that the matrix elements average to the identity, one has a similar structure of probability. However, taking a saddle point would yield a delta function distribution of eigenvalues, whereas in fact the width of the eigenvalue distribution is similar to its average. In muP scaling this issue should go away.

More minor

4. Some experimental data missing (what is the input dimension in the transfer figures)

5. Does the computational cost for saddle point solvers and networks factor in the ability to parallelize networks? If not, this should be clearly stated.

**Questions For Authors:**

See previous comments.

**Relation To Broader Scientific Literature:**

The literature review is Ok.

**Theoretical Claims:**

I looked at the derivation, but did not rederive it. I raised above some concerns about the correctness of the saddle point equations in the NTK setting.


Minor comment. Eq. 60 61 are missing summand indices.

---

> ### Author Rebuttal · Authors · 2025-03-30
>
> We thank the reviewer for their careful reading and thoughtful questions. Below we address the key questions and concerns and hope that the reviewer will be satisfied with our answers and consider an increase in their score.
>
>
> **Result Delivered as Complex Nonlinear Equations**
>
> The theoretical results, while still complicated, are **lower dimensional** than the system that we started with. In general, we went from gradient flow dynamics on $(L-1)N^2 + ND + N$ parameters to a system of equations for $4L+4$ correlation/response functions. Compared to other mean field results which involve *non-Gaussian Monte-carlo averages*, this theory is much simpler since all equations for the order parameters close directly.
>
> That said, the critique that **mean field descriptions can be pretty complicated** (especially for nonlinear dynamics) is a valid concern. However, we were able to extract some useful insights from the equations including
>
> 1. The divergence of response functions with respect to depth $L$ unless the residual branch is scaled as $1/\sqrt{L}$
> 2. Approximate scalings of maximum stable learning rate for mean field and NTK parameterizations (see Figure 7).
> 3. The effect of feature learning on power law convergence rates for power law data (see Figure 6).
> 4. The DMFT equations generally reveal a **buildup** of finite width (finite $\nu$) and finite data (finite $\alpha$) effects over time. Thus the early part of training is much closer to the "clean limit" where $\alpha,\nu \to \infty$ but later in training there are more significant deviations across model sizes or dataset sizes.
>
> We will mention some of the insights that we can extract from the theory more concretely in the main text and Appendex sections.
>
>
>
> **Shift in Optimal LRs**
>
>
> It is true that learning rate transfer for SGD is not perfect in our model, especially when going from **very small widths** to **very large widths**, we point out a few things
>
> 1. The transfer in $\mu$P is much better than for NTK scaling, especially at large widths.
> 2. The success or failure of transfer is captured by our theory (dashed lines of Figure 2).
> 3. In realistic $\mu$P experiments other architectural details that are not included in our model (like Layernorm or Adam) improve the hyperparameter transfer effect in $\mu$P. SGD without layernorm in deep nonlinear networks with $\mu$P looks similar to the "sharp" cutoffs we see in our linear network experiments (see Figure 1 (a)-(b) compared to Figure 2b here https://arxiv.org/abs/2309.16620).
>
>
> **Is the Saddle Point Valid? Do all Eigenvalues Collapse to a Dirac Distribution?**
>
>
> This is a great question and we thank the reviewer for letting us clear this up! TLDR: Yes, the saddle point is valid and no our equations do not indicate a collapse in the density of eigenvalues, but rather capture Marchenko-Pastur like spread in time-constants.
>
> More detail:
>
> 1. We stress that our action is completely independent of $D$ since **none of our vectors of interest carry data indices**. There are only two vectors for each layer $\mathbf h^\ell(t), \mathbf g^\ell(t)\in\mathbb{R}^N$ defined in equation 5. This is the secret sauce of the linear network setting which enables us to take an **exact proportional limit** (note equations 5 and 6 are special for linear networks). Thus the order parameters of interest $C_h^\ell(t,t') = \frac{1}{N} \mathbf h^\ell(t) \cdot \mathbf h^\ell(t')$, etc, also **do not carry data indices**. Thus the number of order parameters is $\mathcal{O}_D(1)$ and we are justified to take the saddle point. This is different than the saddle point of prior works (https://arxiv.org/abs/2205.09653, https://arxiv.org/abs/2304.03408) where the action depends on a number of order parameters which grows with $P$. This is why those works were restricted to $N \to \infty$ limits with $P$ fixed.  **We do not take a saddle point over full $P\times P$ matrices but over correlation and response functions.**
> 2. To illustrate that we are really capturing the full eigenvalue density etc, we can show that our equations for $R_\Delta(t,t')$ can recover the **Marchenko-Pastur law** for a Wishart in the lazy training regime.
>
>
> To illustrate take $\nu \to \infty$, $L=1$ from our DMFT (the flow $\frac{d}{dt} \mathbf v(t) = - \left(\frac{1}{P} \mathbf X^\top \mathbf X \right) \mathbf v(t) + \mathbf j(t)$), where we have that the response $\mathcal H(\omega) \equiv  \int d\tau e^{-i\omega \tau}  \frac{1}{D}  \text{Tr} \frac{\partial \mathbf v(t+\tau)}{\partial \mathbf j(t)^\top}$ satisfies the quadratic equation for the **resolvent of a Wishart matrix**
> \begin{align}
>     \alpha^{-1} (i\omega) \mathcal H(\omega)^2 + (i\omega + 1 - \alpha^{-1}) \mathcal H(\omega) - 1 = 0 .
> \end{align}
> This will give a **distribution over time-constants (eigenvalues)** instead of a single Dirac delta function.
>
> **Missing Experimental Data**
>
> D=1000, we will add.
>
> **Computational cost with Parallelism**
>
> We will include this in the analysis.

---

> > ### Comment · Reviewer_pEUF · 2025-04-03
> >
> > I thank the authors for the detailed reply, which has clarified most of my concerns. Accordingly, I raise my score to 4.

---

### Decision · Program_Chairs · 2025-05-01

**Decision:**

Accept (poster)

**Comment:**

The reviews are mainly positive, and the only negative review has very little concrete criticism. The paper gives a rigorous and complete analysis of the asymptotic behavior of linear DNNs, leading to interesting insights on hyperparameter transfer. I am leaning towards acceptance for this paper.